# Hypoxia-induced SETX links replication stress with the unfolded protein response

Shaliny Ramachandran[1,7], Tiffany S. Ma [1,7], Jon Griffin [2,3], Natalie Ng[1], Iosifina P. Foskolou [1], Ming-Shih Hwang[1], Pedro Victori [1], Wei-Chen Cheng[1], Francesca M. Buffa [1], Katarzyna B. Leszczynska [1,6], Sherif F. El-Khamisy [2,4], Natalia Gromak [5] & Ester M. Hammond [1✉]

Tumour hypoxia is associated with poor patient prognosis and therapy resistance. A unique transcriptional response is initiated by hypoxia which includes the rapid activation of numerous transcription factors in a background of reduced global transcription. Here, we show that the biological response to hypoxia includes the accumulation of R-loops and the induction of the RNA/DNA helicase SETX. In the absence of hypoxia-induced SETX, R-loop levels increase, DNA damage accumulates, and DNA replication rates decrease. Therefore, suggesting that, SETX plays a role in protecting cells from DNA damage induced during transcription in hypoxia. Importantly, we propose that the mechanism of SETX induction in hypoxia is reliant on the PERK/ATF4 arm of the unfolded protein response. These data not only highlight the unique cellular response to hypoxia, which includes both a replication stress-dependent DNA damage response and an unfolded protein response but uncover a novel link between these two distinct pathways.

[1] Department of Oncology, Oxford Institute for Radiation Oncology, University of Oxford, Oxford, UK. [2] Department of Molecular Biology and Biotechnology, Healthy Lifespan and Neuroscience Institute, Firth Court, University of Sheffield, Sheffield, UK. [3] Department of Histopathology, Sheffield Teaching Hospitals NHS Foundation Trust, Sheffield, UK. [4] Institute of Cancer Therapeutics, University of Bradford, Bradford, UK. [5] Sir William Dunn School of Pathology, University of Oxford, Oxford, UK. [6] Present address: Laboratory of Molecular Neurobiology, Neurobiology Center, Nencki Institute of Experimental Biology, Polish Academy of Sciences, Warsaw, Poland. [7] These authors contributed equally: Shaliny Ramachandran, Tiffany S. Ma. ✉email: ester.hammond@oncology.ox.ac.uk

The microenvironment of the majority of solid tumours is characterised by regions of low oxygen (hypoxia), which result from abnormal vasculature and increased oxygen demand. Tumour hypoxia plays a pivotal role in tumorigenesis and has been associated with increased aggressiveness, metastasis, resistance to radiotherapy, and poor patient prognosis[1]. The level of hypoxia associated with radiation resistance (<0.13% $O_2$) is characterised by a rapid induction of replication stress, which is defined as the slowing or stalling of replication forks. We have attributed hypoxia-induced replication stress to decreased nucleotide availability[2–4]. Hypoxia-induced replication stress leads to the induction of an ATR- and ATM-dependent DNA damage response (DDR), although hypoxia-induced replication stress is insufficient to induce ATM-mediated signalling and increased levels of heterochromatic histone marks, notably H3K9me3 are also required[5–7]. Importantly, both hypoxia-induced ATR and ATM activity occur in the absence of detectable DNA damage[8]. Transcriptional stress, defined as global repression of RNA synthesis, has also been identified in the hypoxic conditions which lead to a DDR[9].

R-loops are 3–stranded nucleic acid structures that are linked to transcription and can contribute to replication stress[10]. R-loops are comprised of an RNA/DNA hybrid and a displaced single-stranded DNA (ssDNA), for recent reviews see[11–13]. R-loops form within the genome at actively transcribed genes preferentially occurring at gene promoters and terminators. R-loop homoeostasis is regulated through several pathways including mRNA processing, RNA modifications, DNA topology, single-stranded DNA-binding proteins and by specialised R-loop processing enzymes such as RNase H, RNA/DNA hybrid helicases and DNA nucleases[14–17]. R-loops regulate gene expression but can also lead to genome instability, which is particularly apparent in cells with deficiencies in R-loop processing and binding factors, including for example BRCA1[18–22]. Recently, a role for R-loops has been described in the regulation of RNA polymerase I (Pol I) activity by RNA polymerase II (Pol II)[23]. R-loops have not been studied in the context of hypoxia however, hypoxia initiates a significant change to cellular transcription suggesting there may be an impact on R-loop formation and resolution[9,24]. Despite a reduction in global levels of transcription, a number of transcriptional pathways are induced in hypoxia including the hypoxia-inducible factors (HIF1-3), p53 and those initiated by the unfolded protein response (UPR)[25–27]. The UPR is an ER stress-response pathway that leads to global translation inhibition while upregulating selective genes including those involved in protein folding, redox homoeostasis and protein degradation[28]. In contrast to the DDR, which is activated by ssDNA in hypoxia, the UPR is activated by misfolded proteins. Notably, the DNA repair pathways, including critical R-loop associated factors such as BRCA1 and Rad51, have been shown to be repressed through a variety of mechanisms in response to hypoxia[29–31].

Senataxin (SETX) is a putative RNA/DNA helicase that can facilitate transcription termination and gene transcription of a subset of genes[32–36]. The yeast homologue of SETX, Sen1, was shown to be involved in transcription-coupled repair and protect cells from transcription-associated recombination[37–39]. In addition, Sen1 has been shown to act at the replication fork and facilitate replication at highly transcribed genes[40]. SETX has also been linked to R-loops at transcription–replication collisions and to participate in the antiviral response[39,41]. Mutations in *SETX* have been associated with two neurodegenerative diseases, ataxia with oculomotor apraxia type 2 (AOA2) and amyotrophic lateral sclerosis type 4 (ALS4)[42,43].

The role of R-loops and factors involved in R-loop processing including RNA/DNA helicases have not been investigated in response to hypoxia. We show that in hypoxia, which includes

transcriptional and replication stress, R-loops increase. The majority of the factors investigated associated with R-loops were found to be repressed with the notable exception of SETX, which was induced in hypoxia, prompting us to focus on the role of SETX in hypoxia. We found that the loss of SETX leads to an accumulation of R-loops, and that hypoxia-induced SETX protected cells from replication stress, transcription-dependent DNA damage and apoptosis. Most importantly, hypoxia-induced SETX was dependent on the PERK branch of the UPR, suggestive of a link between the UPR and replication stress in hypoxia.

## Results

**SETX is induced in an oxygen-dependent manner**. Exposure to hypoxia (<0.1% $O_2$) led to the rapid accumulation of RPA foci and pan-nuclear γH2AX in the absence of 53BP1 foci. Importantly, the cells that had pan-nuclear γH2AX also had RPA foci which indicate a replication stress-dependent DDR in the absence of detectable DNA damage (Fig. 1a, b and Supplementary Fig. 1a). This hypoxia-induced DDR occurred in a background of decreased global transcription rates as shown by quantification of 5-ethynyluridine (5′EU) incorporation (Fig. 1c, d and Supplementary Fig. 1b). In these same conditions (<0.1% $O_2$), the transcript levels of a number of R-loop processing factors including RNase H1, RNase H2B, PIF1, DHX9, RTEL1, and AQR, were significantly repressed, in contrast, SETX was upregulated (Fig. 1e). As SETX has not previously been shown to be stress-responsive at the transcriptional level, we verified that hypoxia-mediated induction of SETX was evident in a number of cell lines (Supplementary Fig. 1c–e). SETX protein levels were also upregulated in hypoxia in the cell lines tested (Fig. 1f–h). To determine oxygen dependency, we compared expression levels in <0.1% $O_2$ to 2% $O_2$ and found that SETX was specifically induced at <0.1% $O_2$ (Fig. 1i). Next, we looked for evidence of links between hypoxia and SETX in cancer patients, using The Cancer Genome Atlas (TCGA) colorectal and lung cancers datasets. Here, we compared SETX expression with both, a validated hypoxia signature and with a group of genes that, like SETX, we have previously shown are induced at <0.1% $O_2$ but not 2% $O_2$[25,44]. As we saw no indication of SETX induction in 2% $O_2$, we were not surprised to find that SETX expression did not correlate with the hypoxia signature (Supplementary Fig. 1f). However, in agreement with the SETX induction at <0.1% $O_2$, SETX expression showed a positive correlation with the group of genes we have identified as induced at <0.1% $O_2$ (Fig. 1j)[25]. Therefore, it seems likely that SETX expression is increased in patient samples with tumours experiencing hypoxia-induced replication stress, lending further support to our novel finding that SETX is induced in an oxygen-dependent manner.

**Loss of SETX leads to changes in gene expression in hypoxia**. SETX has been suggested to affect Pol II binding at certain genes and to facilitate transcriptional termination of a number of genes[34,35,45]. Under normoxic conditions SETX knockdown led to a significant decrease in 5′EU incorporation and given that the majority of transcriptional output is rRNA, this agrees with the recent report demonstrating that SETX supports rRNA transcription[23]. In hypoxia where transcription rates are reduced, there was a modest, further decrease in 5′EU incorporation upon SETX depletion (Fig. 2a). A previous study identified changes in the expression of 36 genes in response to SETX depletion, and 60 genes changing in a SETX-dependent manner in response to viral infection[41]. We carried out RNA-seq, which confirmed that SETX was induced in hypoxia by approximately 3-fold and found that SETX depletion affected a small subset of genes (17 increasing and 48 decreasing) in normoxia, and of these genes, 5 were

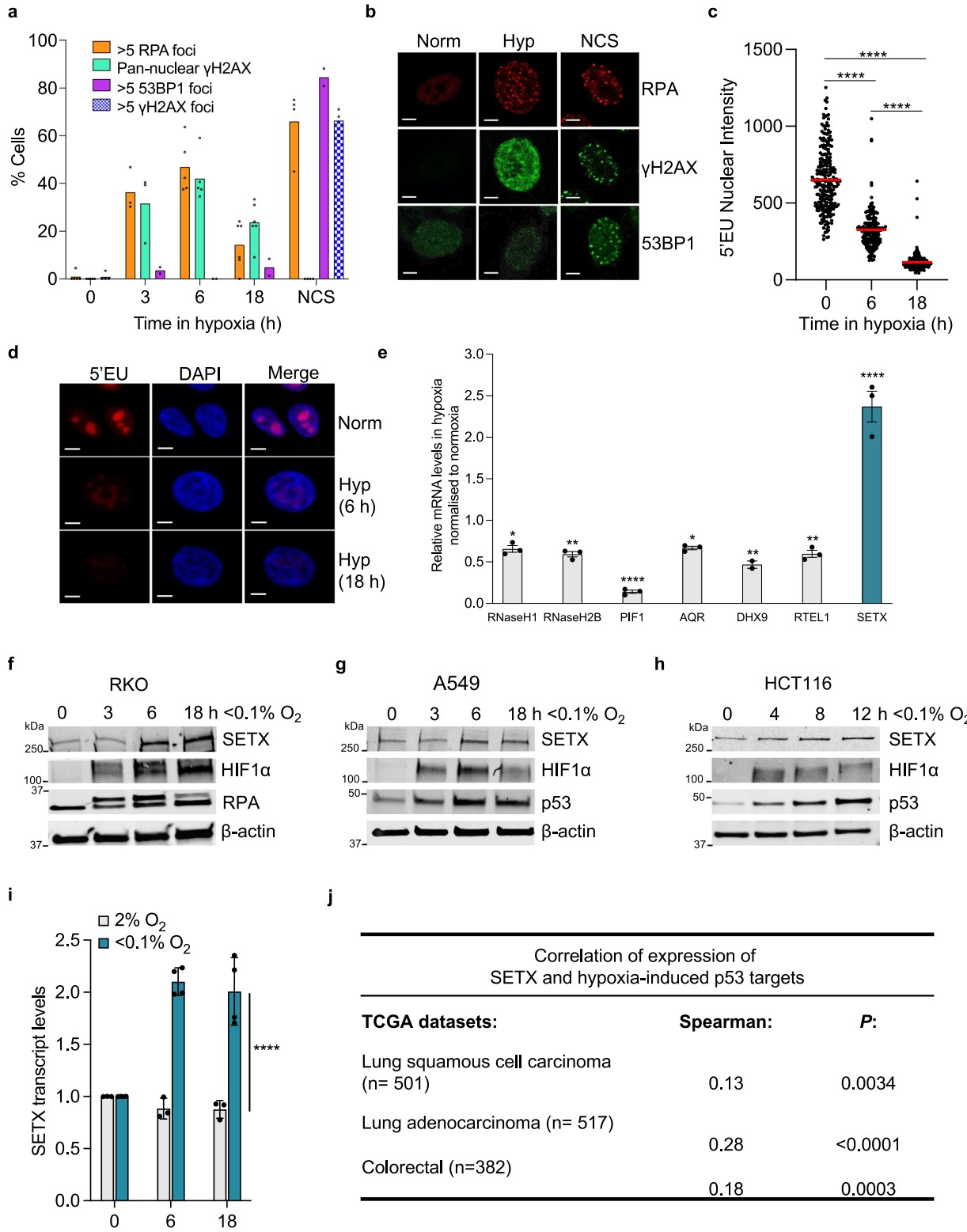

previously identified as regulated in a SETX-dependent manner. In contrast, SETX depletion in hypoxia caused 341 genes to increase and 256 genes to decrease in expression (Fig. 2b). These changes were also validated by RT-qPCR (Fig. 2c). We asked whether the differentially expressed genes shared any common features by gene ontology analysis and found that approximately 20% of genes upregulated in SETX-depleted hypoxic cells were

involved in rRNA processing, nucleolus and ribosome biogenesis (Supplementary Fig. 2a). In addition, hypoxia-induced genes that had increased expression in SETX-depleted cells had a slightly shorter gene length compared to the hypoxia-induced genes that were unaffected or had decreased expression upon SETX depletion (Supplementary Fig. 2b). Differentially expressed genes had similar numbers of exons, G4 quadruple structures or GC per

**Fig. 1 SETX is induced in an oxygen-dependent manner. a** RKO cells were exposed to <0.1% $O_2$ for 0, 3, 6 and 18 h or to NCS (200 ng/μL) for 6 h. Cells were fixed and stained for γH2AX, RPA and 53BP1. The percentage of cells stained with each marker in each condition is shown, individual data points represent the average in separate fields of view. More than 100 cells were counted in each condition. A representative plot (of 2 independent) experiments is shown. **b** Representative images from **a** (scale bar: 5 μm). NCS = neocarzinostatin. **c** RKO cells were exposed to <0.1% $O_2$ for 0, 6 and 18 h with 5′EU (0.5 mM) added for the final 6 h. Representative of 3 independent experiments ($n = 3$). Each dot represents a cell and a minimum of 100 cells was imaged per treatment. One-way ANOVA Tukey's multiple comparisons test was used, ****$p < 0.0001$. **d** Representative images from **c** (scale bar: 5 μm). **e** RKO cells were exposed to <0.1% $O_2$ for 6 h followed by RT-qPCR for the mRNAs indicated relative to the normoxic control. The dashed line indicates normoxic expression levels. One-way ANOVA Dunnett's multiple comparisons test was used: RNase H1 ($p = 0.0162$), RNase H2B ($p = 0.0042$), PIF1 ($p < 0.0001$), AQR ($p = 0.0207$), DHX9 ($p = 0.0010$), RTEL1 ($p = 0.0046$), SETX ($p < 0.0001$). **f** RKO cells were treated for 0, 3, 6, and 18 h of <0.1% $O_2$. The levels of SETX, HIF1α and RPA are shown. β-actin is the loading control. **g** A549 cells were treated for 0, 3, 6, and 18 h of <0.1% $O_2$. The levels of SETX, HIF1α, and p53 are shown. β-actin is the loading control. **h** HCT116 cells at 0, 4, 8, and 12 h of treatment with <0.1% $O_2$. The levels of SETX, HIF1α, p53 and the loading control β-actin, are shown. **i** A549 cells were exposed to hypoxia, 2% $O_2$ or <0.1% $O_2$ for the times indicated and the relative mRNA levels of SETX determined. Two-way ANOVA was used ($p < 0.0001$). **j** To examine *SETX* expression against tumour-associated hypoxia-induced p53 activity (referred to as hypoxia-induced p53 targets), raw data for each sequenced gene were rescaled to set the median equal to 1, and hypoxic p53 targets were determined by quantifying the median expression of 6 p53 target genes associated with hypoxia-induced p53 activity (encoding *BTG2, CYFIP2, INPP5D, KANK3, PHLDA3,* and *SULF2*)[25]. Correlations and statistical significance were determined by calculating Spearman's rho rank correlation coefficient ($r$) and two-tailed *P*-value using Hmisc package in RStudio: Lung squamous cell carcinoma ($p = 0.0034$), lung adenocarcinoma ($p < 0.0001$), Colorectal ($p = 0.0003$). **a–i** Data from three independent experiments ($n = 3$), mean ± standard error of the mean (SEM) are displayed unless otherwise indicated.

cent (Supplementary Fig. 2c–e). Next, we investigated the prevalence of R-loop forming sequences (RLFS) in the genes altered in a SETX-dependent manner using a validated prediction tool[46,47] There was no difference in the RLFS density between up and downregulated genes, however, both gene sets have significantly more RLFS than genes randomly selected from all protein-coding genes (Fig. 2d, e). Together these analyses suggest that SETX has a role in regulating gene expression which is hypoxia specific and is likely linked to a role in R-loop resolution as the affected genes all have high levels of RLFS.

**R-loops accumulate in response to hypoxia.** To determine the impact of loss of SETX in hypoxia on R-loops, we used the S9.6 antibody, which specifically recognises RNA/DNA hybrids. The nuclear intensity of S9.6 increased in response to hypoxia and importantly, this was ablated by treatment with RNase H, an endonuclease that degrades the RNA component of R-loops (Fig. 3a, b). A slot blot using the S9.6 antibody in normoxic and hypoxic conditions also demonstrated an increase in S9.6 staining in hypoxia (Supplementary Fig. 3a). Importantly, an orthogonal assay using the catalytic-dead mutant RNase H1[D210N] to visualise R-loops also determined an accumulation in R-loops in response to hypoxia (Fig. 3c, d)[48,49]. Together, these findings demonstrate that R-loop levels increase in response to hypoxia. To further investigate this surprising finding, we asked how decreasing R-loops in hypoxia through overexpression of RNase H1 would impact the biological response to hypoxia. R-loops can lead to a localised increase in H3K9me2 and to replication stress, both of which are part of the hypoxic response[50,51]. Overexpression of RNase H1 leads to a decrease in H3K9me2 (3-fold) and H3K9me3 (3-fold) in hypoxia but had no significant impact on the levels of these marks in normoxic conditions (Fig. 3e). As expected, exposure to hypoxia led to phosphorylation of RPA and Chk1, confirming replication stress. However, cells overexpressing RNase H1 showed decreased levels of hypoxic RPA and Chk1 phosphorylation, and RPA foci in comparison to the mock-treated cells (Fig. 3f, g). It has been previously shown that the combination of replication stress and a heterochromatic environment lead to an ATM-dependent DDR in hypoxia[5]. Overexpression of RNase H1 led to a decrease in ATM signalling, including reduced KAP1-S824, p53-S15, and γH2AX in hypoxia (Fig. 3h). Together, these data demonstrate that overexpression of RNase H1 significantly impacts both replication stress and the DDR in hypoxia (<0.1% $O_2$) supporting that the biological response is affected by the accumulation of R-loops in hypoxia.

**Loss of SETX leads to DNA damage and decreased viability in hypoxia.** Next, we investigated the role of SETX and found that siRNA mediated depletion of SETX led to an increase in the levels of R-loops in both normoxia and hypoxia, and again, this increase was ablated with RNase H treatment (Fig. 4a and Supplementary Fig. 3b–f). As R-loops can contribute to replication stress, we asked if SETX had a role to play in replication in hypoxia. Depletion of SETX led to a significant increase in stalled replication forks in normoxic conditions (Supplementary Fig. 3g). However, no changes were detected in hypoxic conditions, likely because hypoxia alone induced high levels of stalled forks (Supplementary Fig. 3h). However, SETX depletion also decreased replication rates and this was more significant in hypoxic compared to normoxic conditions (Fig. 4b and Supplementary Fig. 3i, j). Given that unresolved R-loops have been associated with DNA damage, we asked whether depleting SETX would lead to DNA damage in hypoxia. As expected, there was no increase in DNA damage in hypoxia but upon SETX depletion there was a significant increase in DNA damage as determined by increased numbers of cells with 53BP1 foci (Fig. 4c and validated with a second siRNA Supplementary Fig. 4a). To test whether the DNA damage observed in the SETX-depleted hypoxic cells was linked to transcription, we treated cells with dichloro-beta-D-ribofuranosylbenzimidazole (DRB), which inhibits Pol II-mediated transcription. DRB rescued the DNA damage observed in the SETX-depleted hypoxic cells suggesting that the damage observed in SETX-depleted hypoxic cells was transcription-dependent (Fig. 4c). We confirmed that DRB did not rescue DNA damage non-specifically by demonstrating that DRB did not rescue Adriamycin-induced DNA damage (Supplementary Fig. 4b). It should be noted that by inhibiting transcription, DRB led to a decrease in R-loop levels, suggesting that SETX may be protecting the hypoxic cells from co-transcriptional R-loop associated DNA damage (Supplementary Fig. 4c–e). Interestingly, the DNA damage observed in SETX-depleted hypoxic cells occurred in non-S-phase cells, suggesting that the role of SETX in hypoxia is not to mediate transcription–replication conflicts (Fig. 4D). DNA damage, if unrepaired, can lead to increased apoptosis and reduced survival. In hypoxia, SETX depletion led to decreased colony survival when compared to control cells, suggesting that SETX function contributes to hypoxic cell viability (Fig. 4e). SETX depletion also led to a significant increase in apoptosis compared to the control cells specifically in hypoxia and again, this was found to be transcription-dependent as the addition of DRB rescued the apoptosis induced by SETX loss (Fig. 4f and

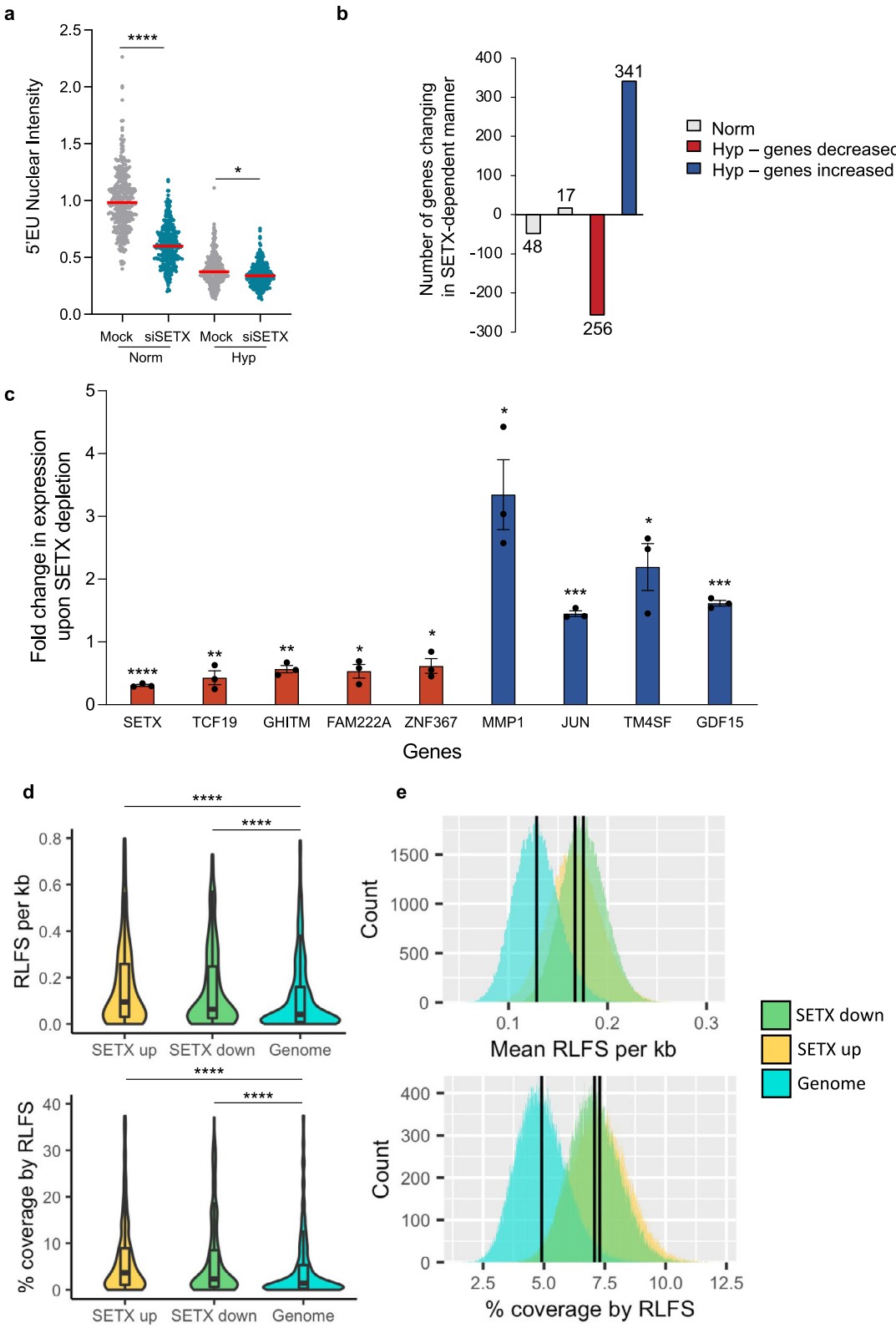

Supplementary Fig. 4f). Taken together, SETX protects hypoxic cells from transcription linked DNA damage and apoptosis.

**SETX upregulation is dependent on the UPR.** SETX has not previously been shown to be stress-responsive, to investigate the mechanism of SETX induction in hypoxia, A549 cells were used as these showed the most robust transcriptional induction of

SETX in hypoxia. Multiple candidate transcription factors are active at the oxygen level (<0.1% $O_2$) shown to induce SETX, including HIF, p53 and those involved in the UPR (ATF3, ATF4, ATF5 and CHOP) (Supplementary Fig. 5a, b). Given that SETX was not induced at 2% $O_2$ where HIF is active, and SETX expression did not correlate with a hypoxia signature, it seemed unlikely that SETX induction was dependent on HIF. However, to

**Fig. 2 Gene expression changes in response to the loss of SETX. a** RKO control cells or SETX-depleted cells were exposed to 21% and <0.1% $O_2$ for 6 h in the presence of 5'EU. Cells were fixed, stained and quantified for 5'EU (0.5 mM) incorporation. Cells treated with siRNA to SETX are shown in green. Data from three independent experiments ($n = 3$) where each dot represents a cell and a minimum of 100 cells were imaged per treatment. One-way ANOVA Tukey's multiple comparisons test was used: ****$p < 0.0001$, *$p = 0.0146$. **b** RNA-seq results comparing RKO cells treated with control siRNA or SETX siRNA in 21% $O_2$ (Norm, grey) and <0.1% $O_2$ (Hyp). Genes with a decreased expression upon SETX depletion in hypoxia shown in red, and genes with an increased expression upon SETX depletion in hypoxia shown in blue. RNA-seq was carried out in 3 biological replicates ($n = 3$). **c** RKO cells were treated with control siRNA or SETX siRNA and exposed to <0.1% $O_2$ for 6 h followed by RT-qPCR. For each gene, relative mRNA levels were determined using 18S as the reference gene. The fold change in expression upon SETX depletion in hypoxia as compared to the control siRNA in hypoxia are shown. Genes in red were repressed in SETX-depleted hypoxic cells, while genes in blue were induced, according to the RNA-seq results. The dotted line indicates no change in expression upon SETX depletion. Data from three independent experiments ($n = 3$), mean ± standard error of the mean (SEM) are displayed. Two-tailed Student's $t$-test was used: SETX ($p < 0.0001$), TCF19 ($p = 0.0066$), GHITM ($p = 0.0017$), FAM222A ($p = 0.0128$), ZNF367 ($p = 0.0305$), MMP1 ($p = 0.0135$), JUN ($p = 0.0006$), TM4SF ($p = 0.0331$), GDF15 ($p = 0.0002$). **d** R-loop forming sequence (RLFS) density measured as RLFS per kb of gene length (upper panel) and % coverage of gene by RLFS (lower panel). The genome plot represents one random sample of all protein-coding genes ($n = 500$). Centre line = median; upper and lower bounds of box = 75th and 25th percentiles (3rd and 1st quartiles); upper and lower bounds of the whiskers = 1.5 times the interquartile range from the corresponding 'hinge' where the line originates from the box; minima and maxima are given by the top and bottom of the overlaid violin plot. $n = 500$ represents the number of protein-coding genes randomly selected to create the null genome distribution shown in light blue. The SETX up and SETX down violin plots used all of the differentially expressed in the corresponding group. Two-tailed Student's $t$-test was used: $p$-value = **** = $2.2 \times 10^{-16}$. **e** Resampling of RLFS densities used to calculate $p$-values shown in **d**. Black vertical lines represent the mean of each distribution.

formally test this, we depleted HIF1α and asked whether this affected SETX induction in hypoxia. As expected SETX induction in hypoxia was not affected by HIF1α siRNA in contrast to the control gene VEGF, confirming that SETX is not induced by HIF1α in hypoxia (Supplementary Fig. 5c, d). We also demonstrated that SETX expression was not induced with the hypoxia mimetic desferrioxamine (DFO) and 2% $O_2$, which lead to HIF stabilisation (Supplementary Fig. 5e, f). Next, we found that SETX induction in hypoxia was not affected by p53 siRNA, suggesting that p53 signalling does not contribute to SETX induction in hypoxia (Fig. 5a and Supplementary Fig. 5g). In support of this conclusion, we and others have found no evidence of a functional p53 response element in the SETX promoter[52]. Next, we investigated whether SETX induction could be linked to the UPR. Predicted binding sites were identified for the UPR transcription factor ATF4 in the GeneHancer identifier GH09J132349 corresponding to the *SETX* promoter region. We depleted ATF4 using siRNA and demonstrated that ATF4 is responsible for the induction of SETX in hypoxia (Fig. 5b and Supplementary Fig. 6a). The transcript levels of the ATF4 target CHOP were tested in parallel to confirm ATF4 knockdown (Supplementary Fig. 6b). ChIP was used to determine direct ATF4-mediated transactivation of SETX and demonstrated that ATF4 was enriched at the SETX promoter specifically in hypoxia (Fig. 5c). ATF4 enrichment at the CHOP gene was used as a positive control (Supplementary Fig. 6c). To further investigate whether SETX is a UPR target, we exposed cells to a range of stresses including tunicamycin and thapsigargin, both of which are known to induce the UPR, and found that both led to a significant increase in SETX mRNA similar to the induction observed in hypoxia (<0.1% $O_2$) (Fig. 5d, Supplementary Fig. 6d). Consistent with this, SETX was found to be upregulated in published RNA sequencing datasets upon tunicamycin or thapsigargin treatment[53,54]. We considered that SETX might be responsive to replication stress and that tunicamycin and thapsigargin could induce replication stress. However, hydroxyurea (Hu) did not induce SETX (Fig. 5d) and we saw no evidence of replication stress-induced in response to either thapsigargin or tunicamycin (Fig. 5e and Supplementary Fig. 6e, f). The data presented so far suggest that the hypoxia-induced UPR reduces replication stress through PERK/ATF4-dependent induction of SETX. To interrogate the relationship between the UPR and replication stress we made use of agents which, unlike hypoxia, induce either a UPR or replication stress but not both (Fig. 5e). Cells were treated with Hu, to induce

replication stress, with or without thapsigargin and the number of cells with RPA foci determined in each condition. The number of cells with Hu-induced RPA foci were significantly reduced by thapsigargin treatment (Fig. 5f, g). Importantly, we determined that this was not due to decreased expression of RPA which could have occurred due to reduced translation rates induced by the UPR (Supplementary Fig. 6g). Signalling to ATF4 in hypoxia is primarily mediated by the protein kinase R (PKR)-like endoplasmic reticulum kinase (PERK)[55,56] which led us to the hypothesis that the reduction in Hu-induced replication stress observed in the presence of thapsigargin could be PERK-mediated. In support of this hypothesis, we demonstrated again that adding thapsigargin to Hu-treated cells reduced RPA phosphorylation (Fig. 5h) but that this effect was abrogated if a PERK inhibitor GSK2606414 (PERKi) was also included (Fig. 5i).

**The UPR is linked with replication stress in hypoxia**. As expected, depleting PERK using siRNA abrogated the hypoxia-mediated induction of SETX (Fig. 6a and Supplementary Fig. 6h). In addition, PERKi led to a significant decrease in SETX mRNA levels in hypoxia as compared to the control cells (Supplementary Fig. 6i, j). We also verified that pre-treating cells with PERKi abrogated thapsigargin-mediated SETX induction, again demonstrating that SETX induction during the UPR is downstream of PERK (Fig. 6b and Supplementary Fig. 6k).

Since SETX depletion led to an increase in R-loops, we hypothesised that PERK depletion would have a similar effect. To test our hypothesis, we used both S9.6 and RNaseH1^D210N to determine changes in R-loop levels with siRNA to PERK or PERKi. siRNA mediated PERK depletion and PERKi led to an increase in R-loop levels in hypoxia using both detection methods suggesting that, like SETX, PERK signalling negatively regulates R-loops in hypoxia (Fig. 6c and Supplementary Fig. 7a–e). As SETX depletion led to an increase in transcription-dependent DNA damage, we asked whether PERK inhibition showed a similar effect. Akin to SETX depletion, PERK inhibition led to an accumulation of DNA damage which was more significant under hypoxic conditions and dependent on transcription (Fig. 6d). Together, these data suggest a role for PERK in reducing replication stress and DNA damage in hypoxia. In agreement with this conclusion, when PERK was inhibited in hypoxia we saw an increase in both phosphorylations of p53 and RPA indicating an exacerbation of the DDR and replication stress (Fig. 6e). To investigate further, we determined the number of

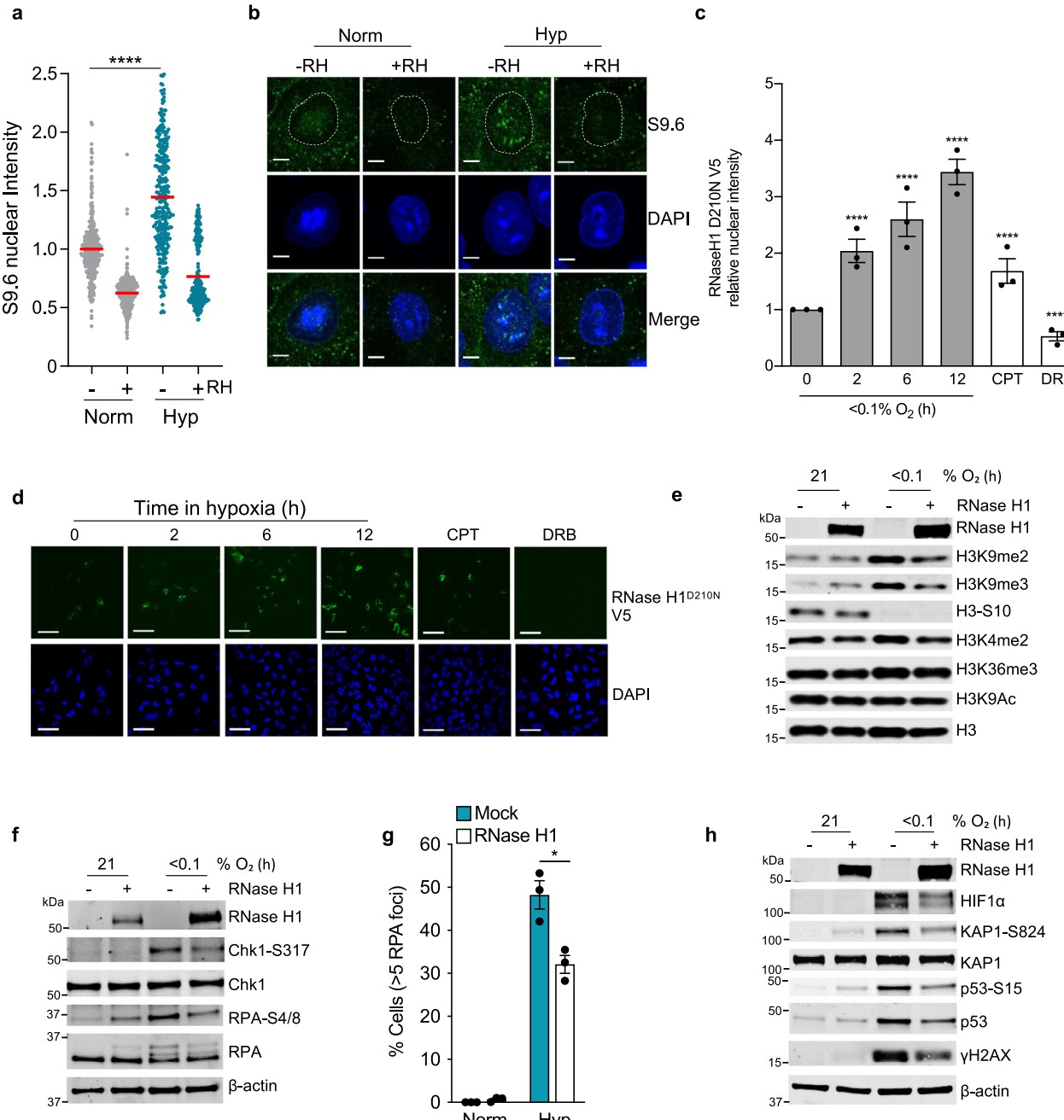

**Fig. 3 Accumulation of R-loops in hypoxia (<0.1% O2). a** A549 cells were exposed to 21% $O_2$ or <0.1% $O_2$ (shown in green) for 12 h, fixed and stained with the S9.6 antibody and DAPI. Where indicated, coverslips were treated with RNase H prior to staining. Data from three independent experiments ($n =$ 3) where each dot represents a cell and a minimum of 100 cells were imaged per treatment. One-way ANOVA Tukey's multiple comparisons test was used, ****$p < 0.0001$. **b** Representative images from **a** (scale bar: 5 μm). A dashed white line indicates a nuclear outline based on a DAPI stain that was used to measure the nuclear S9.6 intensity. **c** A549 cells were transfected with RNase HI[D210N]V5 and exposed to hypoxia (<0.1% $O_2$), CPT (10 μM, 20 min), or DRB (100 μM, 1 h). Staining for V5 was then carried out and the nuclear intensity determined. The two-tailed Student's $t$-test was used: ****$p <$ 0.0001. **d** Representative images from part **c** are shown (scale bar: 30 μm). V5 staining is green, DAPI to show the nucleus is blue. **e** HCT116 cells were transfected with mock or RNase H1 plasmid and exposed to 21% $O_2$ or <0.1% $O_2$ for 6 h. RNase H1 overexpression was confirmed by western blot analysis. Changes in chromatin modifications H3K9me2, H3K9me3, H3-S10, H3K4me2, H3K36me3, and H3K9Ac are shown. Total H3 was included as a loading control. **f** HCT116 cells were treated as in part **e** (<0.1% $O_2$ for 6 h). The levels of RPA-S4/8 and Chk1-S317 are shown. RPA, Chk1, and β-actin controls were included. **g** HCT116 cells transfected with mock or RNase H1 plasmid and exposed to 21% $O_2$ (Norm) or <0.1% $O_2$ (Hyp) for 6 h were stained for RPA and the % of cells with >5 RPA foci determined. Two-tailed Student's $t$-test was used: $p = 0.0142$. **h** Western blotting was carried out on cells treated as in part **g**. ATM activation was detected using KAP1-S824, p53-S15, and γH2AX. Total KAP1, p53, H3 and β-actin control westerns were included. **a–h** Data from three independent experiments ($n = 3$), mean ± standard error of the mean (SEM) are displayed unless otherwise indicated.

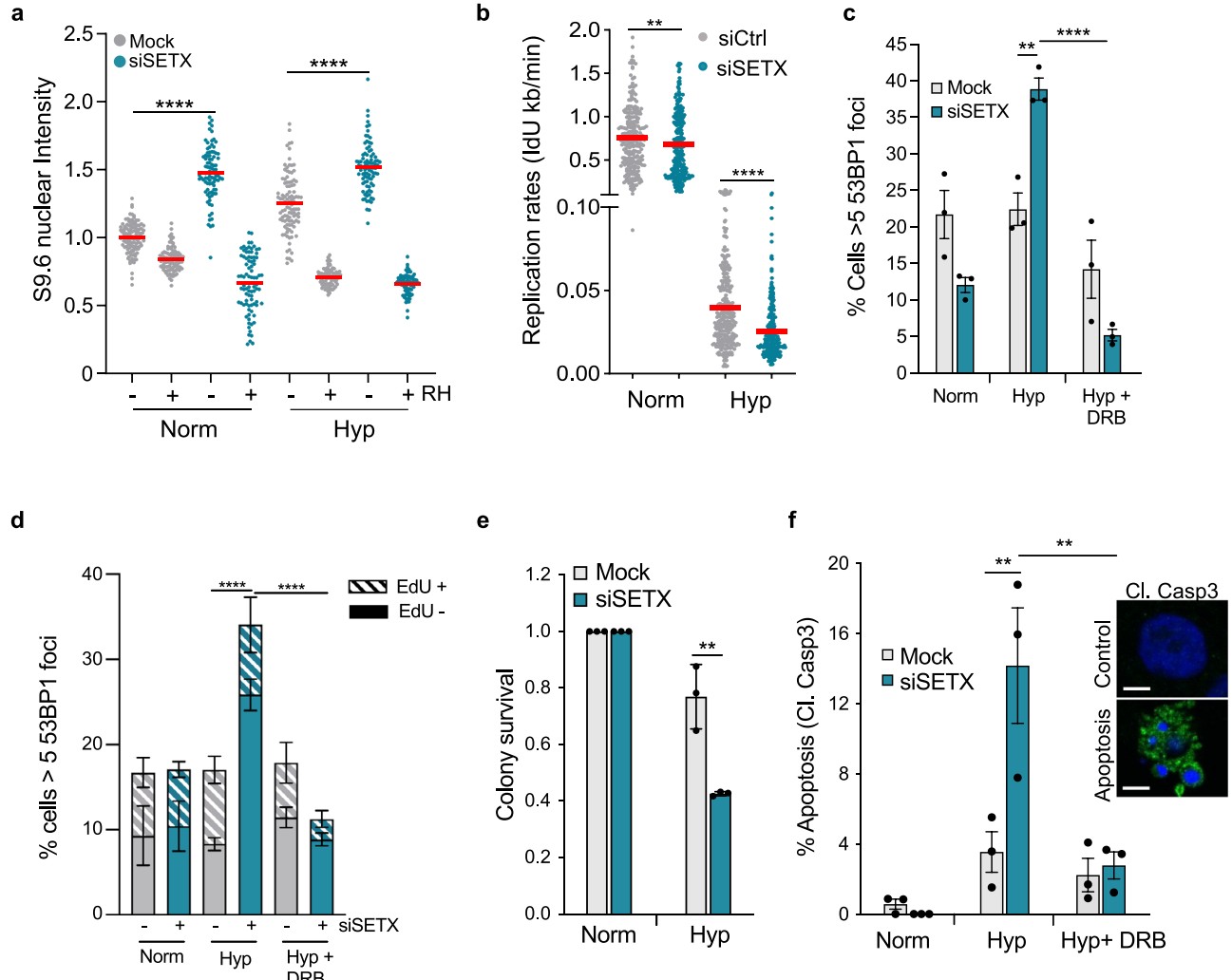

**Fig. 4 SETX reduces R-loops and protects from DNA damage and cell death in hypoxia. a** A549 cells that were either mock-transfected (grey) or transfected with SETX siRNA (green), were exposed to 21% $O_2$ or <0.1% $O_2$ for 12 h. Cells were fixed and stained with the S9.6 antibody and DAPI. Where indicated, coverslips were treated with RNase H prior to staining. Data from one independent experiment ($n = 1$) where each dot represents a cell and a minimum of 100 cells were imaged per treatment. **b** Replication rates, as determined by IdU incorporation rates, of RKO cells transfected with control siRNA versus SETX siRNA exposed to 21% $O_2$ (Norm) or <0.1% $O_2$ (Hyp) for 6 h. Data from three independent experiments ($n = 3$) where each dot represents a fibre and a minimum of 100 fibres were analysed per treatment. Two-tailed Mann–Whitney $U$ test was used: ****$p < 0.0001$, *$p = 0.0068$. **c** Percentage of cells displaying 53BP1 foci (>5 per nucleus) in RKO cells with either mock-treated or treated with SETX siRNA. Cells were exposed to 21% $O_2$ (Norm), 6 h of <0.1% $O_2$ (Hyp) or pre-treated with DRB (100 μM) followed by 6 h of <0.1% $O_2$ (Hyp + DRB). One-way ANOVA Tukey's multiple comparisons test was used: $^{ns}p = 0.1240$, **$p = 0.0046$, ****$p < 0.0001$. **d** A549 cells that were either mock-transfected (grey) or transfected with SETX siRNA (green) were pulsed with EdU (10 μM) for 1 h then and exposed to 21% $O_2$ (Norm), <0.1% $O_2$ (Hyp) for 6 h or treated with DRB (100 μM) in <0.1% $O_2$ for 6 h (Hyp + DRB). The percentage of cells displaying 53BP1 (>5 per nucleus) is shown. Two-tailed Student's $t$-test was used: ****$p \leq 0.0001$. **e** Colony survival of A549 cells with either mock treatment or SETX siRNA after 0 and 18 h of hypoxia (Hyp) (<0.1% $O_2$) treatment. Two-tailed Student's $t$-test was used: $p = 0.0065$. **f** A549 cells were mock-treated or treated with SETX siRNA and exposed to 21% $O_2$ (Norm), <0.1% $O_2$ (Hyp) for 24 h or pre-treated with DRB (100 μM) followed by 24 h of <0.1% $O_2$ (Hyp + DRB). The percentage of cells positive for cleaved caspase 3 (Cl. Casp3) is shown. Representative images of a normal and apoptotic cell are included (scale bar: 5 μm). One-way ANOVA Tukey's multiple comparisons test was used: Hyp Mock versus Hyp siSETX (**$p = 0.0035$), Hyp siSETX versus Hyp+ DRB siSETX (**$p = 0.0020$). **a–f** Data from three independent experiments ($n = 3$), mean ± SEM are displayed unless otherwise indicated. Source data for **d** is provided as a Source Data file.

cells experiencing replication stress as measured by those with RPA foci when PERK was lost and found that the percentage of cells experiencing replication stress in hypoxia did not change with PERK depletion (Supplementary Fig. 7f). This was not entirely surprising as cells exposed to hypoxia (<0.1% $O_2$) demonstrate high levels of replication stress due to reduced dNTP levels. However, it was apparent that the cells with reduced PERK activity had increased numbers of RPA foci per cell in hypoxia

suggesting again that hypoxia cells with reduced PERK activity have increased levels of replication stress (Fig. 6f, g and Supplementary Fig. 7g). Finally, we tested whether PERK inhibition led to increased replication stress in hypoxia using the DNA fibre assay. Inhibition of PERK led to decreased replication rates in both normoxia and hypoxia, suggesting that PERK signalling acts to alleviate replication stress (Fig. 6h). Taken together, our data suggest that in hypoxia, the PERK branch of

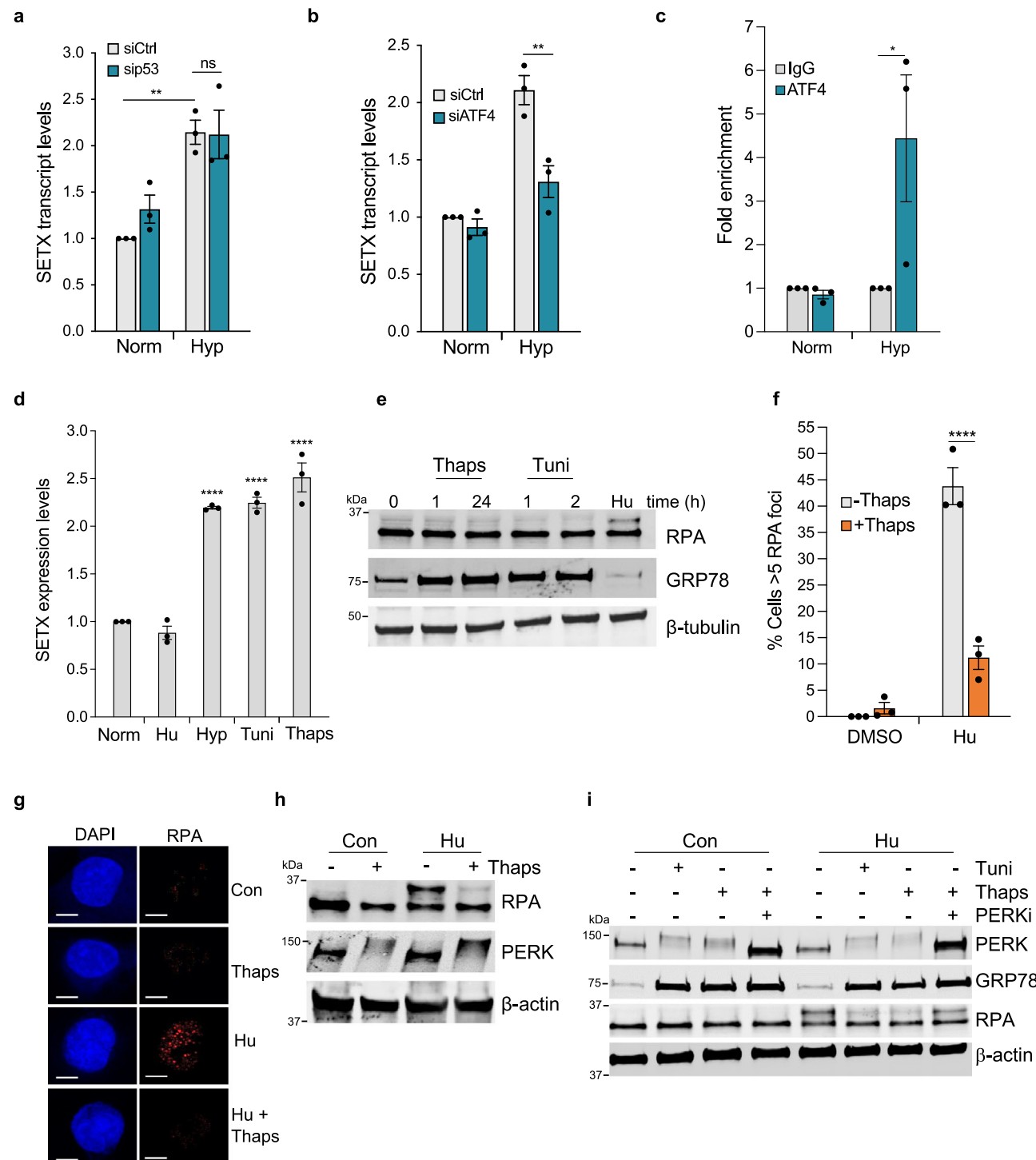

the UPR signals through SETX and potentially other unidentified factors to reduce R-loops, replication stress, DNA damage and apoptosis (Fig. 6i).

## Discussion

Our study suggests that R-loops accumulate in response to hypoxia and that this biological response includes induction of the RNA/DNA helicase SETX. Hypoxia-induced SETX reduces replication stress and protects hypoxic cells from transcription-associated DNA damage and apoptosis. Importantly, SETX induction is dependent on the PERK branch of the UPR. PERK

inhibition increased R-loops and replication stress, uncovering an entirely novel aspect of PERK function in hypoxia.

The accumulation of R-loops in hypoxia may act as a signal of transcriptional stress to the cell, leading to a further cascade of stress-response signalling pathways. R-loops have previously been shown to activate non-canonical ATM signalling through spliceosomal displacement[57]. Our data show that by reducing R-loops through the overexpression of RNase H1 in hypoxia, both the DDR and replication stress responses are alleviated. The reduction in replication stress in hypoxia upon RNase H1 overexpression could be due to less R-loops forming barriers to replication fork progression[58]. Together, these data highlight

**Fig. 5 SETX upregulation is dependent on the PERK branch of the UPR. a** A549 cells were treated with either control siRNA or p53 siRNA and exposed to 21% $O_2$ (Norm) or <0.1% $O_2$ (Hyp) for 6 h. SETX mRNA levels were determined by RT-qPCR. One-way ANOVA Tukey's multiple comparisons test was used: $^{ns}p = 0.9996$, $^{**}p = 0.0050$. **b** A549 cells were treated with either control siRNA or ATF4 siRNA and exposed to 21% $O_2$ (Norm) or <0.1% $O_2$ (Hyp) for 16 h. SETX mRNA levels were determined by RT-qPCR. One-way ANOVA Tukey's multiple comparisons test was used: $^{**}p = 0.0023$. **c** A549 cells were exposed to 21% $O_2$ (Norm) or <0.1% $O_2$ (Hyp) for 4 h, and ChIP-qPCR experiments were performed to determine fold enrichment of ATF4 at the SETX promoter relative to IgG control. One-way ANOVA Tukey's multiple comparisons test was used: $^{*}p = 0.0416$. **d** SETX mRNA levels in non-treated A549 cells (Norm), Hu (1 mM), <0.1% $O_2$ (Hyp), tunicamycin (Tuni, 5 µg/mL) or thapsigargin (Thaps, 2 µM) for 16 h. One-way ANOVA Dunnett's multiple comparisons test was used: $^{****}p < 0.0001$. **e** A549 cells were exposed to Thapsigargin (Thaps) (2 µM) or Tunicamycin (Tuni) (5 µg/mL) for the times indicated, or to Hu (2 mM, 8 h) followed by western blotting. **f** Percentage of cells displaying RPA foci (>5 per nucleus) in A549 control or Hu (2 mM) treated cells that were also treated with or without Thaps (2 µM) for 24 h. The two-tailed Student's t-test was used: $^{****}p < 0.0001$. **g** Representative images from **f** (scale bar: 5 µm). **h** A549 control or Hu (2 mM) treated cells were also treated with or without Thaps (2 µM) for 24 h followed by western blotting. β-actin was used as the loading control. **i** A549 control or Hu (2 mM) treated cells were also treated with DMSO, Tuni (5 µg/mL), Thaps (2 µM) or Thaps (2 µM) and PERKi (10 µM) for 24 h followed by western blotting. β-actin was used as the loading control. **a–i** Data from three independent experiments (n = 3), mean ± SEM are displayed unless otherwise indicated.

transcriptional stress as an additional signal to nucleotide depletion in the induction of ATM/ATR signalling in hypoxia. R-loops are therefore implicated as making a significant contribution to the biological response to hypoxia.

While the DDR and UPR are both active in hypoxia, the DDR is restricted to S phase cells[5]. The UPR/PERK dependency of SETX induction ensures that cells in all phases of the cell cycle increase SETX levels in hypoxia. Combined with our findings using an inhibitor of transcription, this suggests a role for SETX in resolving co-transcriptional R-loops. We hypothesise that these hypoxia-induced R-loops occur at stress-responsive genes which are induced in response to hypoxia, for example, HIF targets. Previous reports have demonstrated that an accumulation of R-loops can lead to an increase in DNA damage. However, hypoxia does not lead to an accumulation of DNA damage even after extended periods of time, suggesting that the hypoxia-induced R-loops are distinct to those induced by other genotoxic stresses such as camptothecin and G-quadruplex ligands[59,60]. The DNA damage we observed in SETX-depleted hypoxic cells is transcription-dependent, supporting the hypothesis that the role of hypoxia-induced SETX is to resolve the co-transcriptional R-loops that are prone to DNA damage and therefore reduce genomic instability.

Lending support to our finding that SETX is induced by the UPR in hypoxia, RNA-sequencing showed that SETX knockdown led to increased transcription of rRNA processing, nucleolar and ribosome biogenesis genes in hypoxia. Given that the UPR is known to inhibit rRNA transcription and protein translation, it is possible that one of the mechanisms that contribute to these functions of the UPR is through the induction of SETX. A role for SETX in hypoxia-induced UPR-mediated repression of rRNA and translation warrants further investigation.

Interestingly, a previous report showed that in response to a short exposure (1 h) of thapsigargin or DTT, PERK inhibits replication fork progression, and origin firing through a Claspin/Chk1 pathway[61]. While it is challenging to interpret these seemingly contradictory findings with those presented here, this does further support a role for the UPR in replication stress. Differences likely arise from the use of chemically induced UPR signalling and the hypoxia-induced UPR and the timeframe of the experiments carried out[62].

SETX has been identified as a potential tumour suppressor[63] and in support of this, SETX expression was found to be reduced in a number of cancers compared to normal tissue controls[43]. However, AOA2 patients do not present with increased cancer susceptibility suggesting a level of redundancy with other R-loop resolution mechanisms exists to protect genome stability. In this study, SETX depletion increased R-loops, replication stress, DNA damage and apoptosis in hypoxia suggesting that SETX may

promote hypoxic tumour growth, and hence be a potential therapeutic target. Interestingly, SETX was recently identified as a gene whose knockout increases sensitivity to cisplatin in a set of genome-wide CRISPR-Cas9 screens[64]. It is possible that in the hypoxic microenvironment in which multiple pathways including DNA repair and potentially R-loop resolution factors are repressed, loss of SETX function would not be readily compensated by other mechanisms in a manner similar to its deficiency in the cerebellum. In addition, this study demonstrates that inhibiting PERK leads to increased R-loops, DNA damage and replication stress in hypoxia, and therefore PERK could also be a potential therapeutic target. Targeting PERK in cancer has been controversial as PERK has demonstrated both tumours promoting and suppressive activities depending on the type of cancer and intensity of stress[65]. However, in hypoxia, PERK supports tumour survival and growth, therefore suggesting PERK as a potential therapeutic target specifically for hypoxic tumours[66,67]. Since PERK inhibition also increased replication stress in hypoxic conditions, we postulate that using PERK inhibitors in conjunction with replication stress inducers such as ATR inhibitors may be beneficial in solid tumours.

## Methods

**Cell lines and reagents.** RKO, HCT116 and A549 (ATCC) were grown in DMEM supplemented with 10% FBS and cultured in humidified incubators at 37 °C and 5% $CO_2$. Cells were routinely tested for mycoplasma and found to be negative. For plasmid transfections, JetPrime (Polyplus transfection) was used. ppyCAG_RNaseH1_WT (#111906) and ppyCAG_RNaseH1_D210N (#111904) were supplied by Addgene. siRNA transfections were performed using Dharma-FECT-1 (Thermo-Fisher Scientific), sequences are available in the SI. siSETX-A siRNA was used for Figs. 2d–f, 3b and Supplementary Figs. 2D–G, 3, 4, 5A, G. siSETX-B siRNA was used for Figs. 2c, 3a, c–f and Supplementary Figs. 2B, C and 5F. AllStars negative control siRNA (Qiagen) and ON-TARGETplus non-targeting pool (Dharmacon) were used as control siRNA. Specific siRNA sequences are shown in Supplementary Table S2. Drugs used were the radiomimetic neocarzinostatin (NCS), DRB, Camptothecin (CPT), Hydroxyurea, hypoxia mimetic DFO, Adriamycin (Sigma), tunicamycin and thapsigargin (MP Biomedicals), and PERK inhibitor GSK2606414 (Calbiochem). Irradiations were performed using a Gamma Service GSR D1 Cs137 irradiator.

**Hypoxia treatment.** Bactron II and BactronEZ anaerobic chambers (Shel Lab) were used for <0.1% $O_2$. An M35 hypoxia work station (Don Whitley Ltd) was used for 2% $O_2$. All experiments at <0.1% $O_2$ were plated in glass dishes, except for colony survival assays, which were plated in 6 well plates. All hypoxic treatments were harvested within the chamber using equilibrated buffers. Oxygen tensions were confirmed with the Oxylite probe (Oxford Optronix).

**Immunoblotting.** Cells were harvested in UTB lysis buffer (9 M Urea, 75 mM Tris-HCl pH 7.5, 0.15 M β-mercaptoethanol). Blots were visualised using the Odyssey Infrared imaging system. Antibodies used were SETX, KAP1 (Bethyl); GRP78, HIF1α, (BD Biosciences); ATM-S1981, RPA-S4/8, β-Tubulin (Abcam); ATM, RPA, KAP1-S824, p53-S15, H3K9me2, H3K4me2, H3K9Ac, H3-S10, H3, H3K36me3, Chk1-S317, PERK (Cell Signaling); p53, β-actin, RNase H1, Chk1 (Santa Cruz); and H3K9me3, γH2AX, H2AX (Millipore).

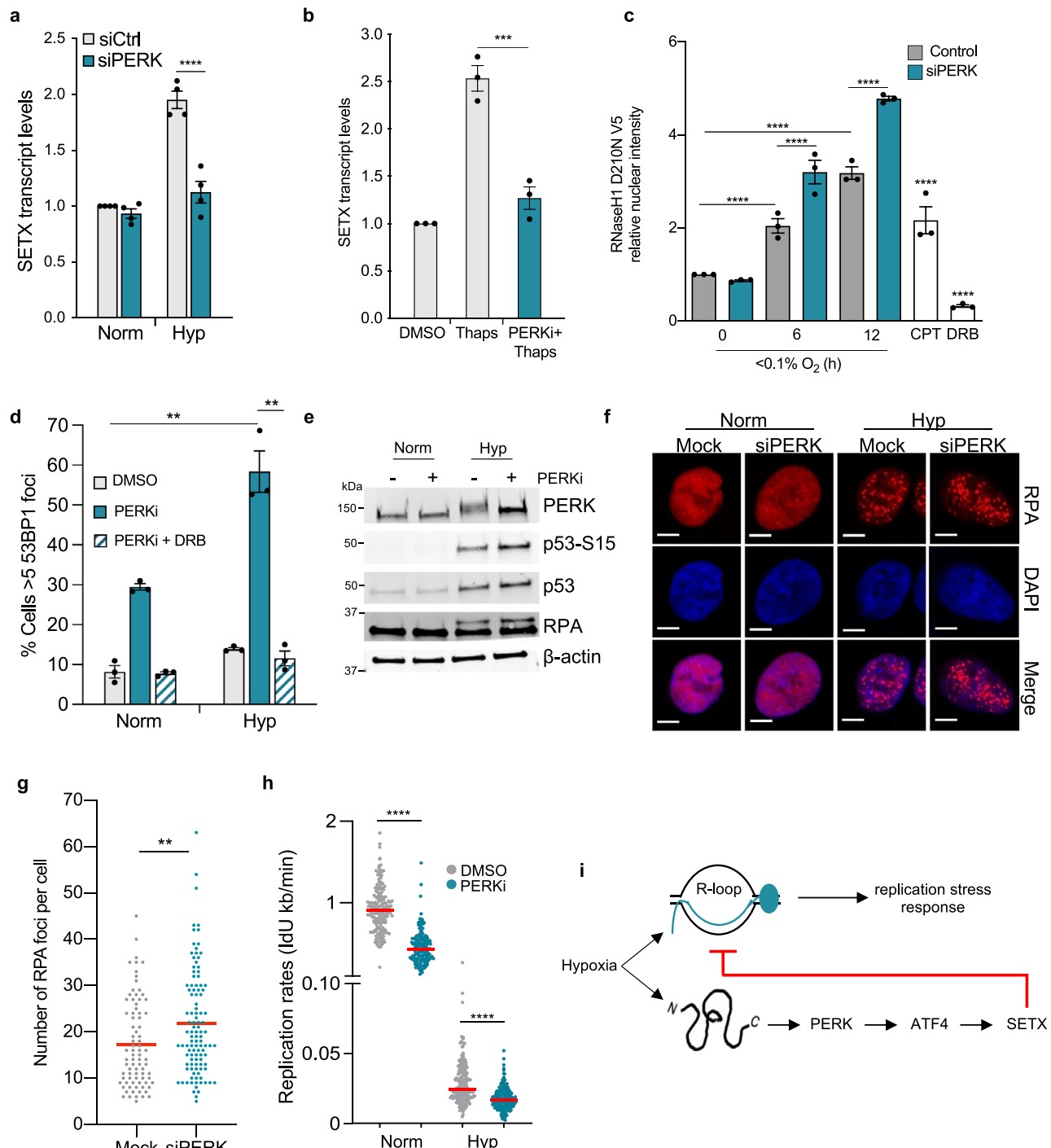

**RT-qPCR**. RNA was extracted using TRI reagent (Sigma), treated with DNase I (NEB) and cDNA prepared using the Verso cDNA synthesis kit (Thermo-Scientific). SYBR Green PCR master mix (Applied Biosystems) and 7500 FAST Real-Time PCR machine (Applied Biosystems) were used. The ΔΔCt method was used to determine relative mRNA fold change with 18S as the reference gene. Primer sequences can be found in Table S1.

**Chromatin immunoprecipitation (ChIP) - qPCR**. Treated cells were fixed with 1% formaldehyde and quenched in 125 mM glycine. Cells were lysed with SDS lysis buffer (0.5% SDS, 10 mM EDTA, 50 mM tris pH 8.1). After sonication with Diagenode Bioruptor, samples were precleared with protein A agarose beads and subsequently incubated with ATF4 antibody (Cell Signaling) overnight.

Antibody–antigen complexes were then pulled down with agarose beads and washed with the following buffers—low salt wash buffer (0.1% SDS, 1% Triton-X 100, 2 mM EDTA, 20 mM Tris pH 8.1, 150 mM NaCl), high salt wash (0.1% SDS, 1% Triton-X 100, 2 mM EDTA, 20 mM Tris pH 8.1, 500 mM NaCl), LiCl wash buffer (1% Igepal, 10 mM Tris pH 8.1, 250 mM LiCl, 1 mM EDTA, 1% sodium deoxycholate), TE wash buffer (10 mM Tris pH 8.0, 1 mM EDTA). DNA was eluted with 0.1 M NaHCO₃ and 1% SDS, cross-linking reversed and treated with proteinase K. DNA was purified with Qiagen PCR purification kits and qPCR was carried out using the following primers: SETX For: CTCAGGTGTCT-CAGCGGATG, SETX Rev: CGCATTGTTCGCAAGACCTA, CHOP For: AAG AGGCTCACGACCGACTA and CHOP Rev: ATGATGCAATGTTTGGCAAC. The amount of immunoprecipitated material was calculated as the fold enrichment over IgG control.

**Fig. 6 Hypoxia links the UPR with replication stress. a** A549 cells were treated with either control siRNA or PERK siRNA and exposed to 21% $O_2$ (Norm) or <0.1% $O_2$ (Hyp) for 16 h. SETX mRNA levels were determined by RT-qPCR. One-way ANOVA Tukey's multiple comparisons test was used: ****$p <$ 0.0001. **b** A549 cells were treated with or without PERKi (10 μM) and exposed to DMSO or Thaps (2 μM) for 6 h. SETX mRNA levels were determined by RT-qPCR. One-way ANOVA Tukey's multiple comparisons test was used: ***$p = 0.0003$. **c** A549 cells were either mock-treated or treated with PERK siRNA, transfected with RNase HI$^{D210N}$V5 and exposed to hypoxia (<0.1% $O_2$), CPT (10 μM, 20 min) or DRB (100 μM, 1 h). Staining for V5 was then carried out and the nuclear intensity determined. The two-tailed Student's $t$-test was used: ****$p < 0.0001$. **d** The percentage of cells displaying 53BP1 foci (>5 per nucleus) as determined by immunofluorescence, in A549 cells exposed to 21% $O_2$ (Norm) or 6 h of <0.1% $O_2$ (Hyp). Cells were treated with DMSO, PERKi (10 μM), or DRB (100 μM) and PERKi (10 μM) (PERKi + DRB). The two-tailed Student's $t$-test was used: Norm PERKi versus Hyp PERKi (**$p$ = 0.0067), Hyp + PERKi versus Hyp+ PERKi + DRB (**$p = 0.0068$). **e** A549 cells were treated with DMSO or PERKi (10 μM) and exposed to normoxia or hypoxia (<0.1% $O_2$) for 6 h followed by western blotting as indicated. PERK inhibition was confirmed by the absence in the electrophoretic mobility shift of PERK on the western blot. **f** A549 cells with either mock treatment or PERK siRNA were exposed to hypoxia (<0.1% $O_2$) for 6 h, cells were fixed and stained for RPA by immunofluorescence assays. Representative images are shown (scale bar: 5 μm). **g** Quantification of the number of RPA foci per cell from **f**. Treatment with siRNA PERK is indicated in green. Data from two independent experiments ($n = 2$) where each dot represents a cell and a minimum of 100 cells were imaged per treatment. Two-tailed Mann–Whitney U test was used: **$p = 0.0024$. **h** Replication rates, as determined by IdU incorporation rates, of A549 cells treated with DMSO or PERKi (10 μM) and exposed to 21% $O_2$ (Norm) or <0.1% $O_2$ (Hyp) for 6 h. Data from two independent experiments ($n = 2$) where each dot represents a fibre and a minimum of 100 fibres were analysed per treatment. Two-tailed Mann–Whitney U test was used: ****$p < 0.0001$. **i** Hypoxia leads to the activation of the UPR and replication stress. In addition to a shortage of nucleotides, R-loops accumulate in hypoxia and contribute to replication stress. The PERK branch of the UPR signals through ATF4 to increase SETX expression in hypoxia. Hypoxia-induced SETX reduces R-loop accumulation and subsequent replication stress-mediated signalling. **a–h** Data from three independent experiments ($n = 3$), mean ± SEM are displayed unless otherwise indicated.

**Immunofluorescence and confocal microscopy**. Cells were fixed in 4% (w/v) paraformaldehyde in PBS, permeabilised in 1% Triton-X 100 in PBS and blocked with 2% (w/v) BSA in 0.1% Tween-20 in PBS. Primary antibodies 53BP1 (Novus Biologicals), RPA (Cell Signaling) and cleaved caspase 3 (Cell Signaling) were used. For 5′EU staining, cells were incubated with 5′ethynyluridine (EU) (0.5 mM) (Jena Bioscience) and Click-iT Alexa Fluor 647 labelling kit (ThermoFisher) was used to measure nascent transcription in the entire nucleus. For EdU staining, cells were incubated with EdU (10 μM) and Click-iT Alexa Fluor 647 labelling kit (ThermoFisher) was used to measure replication. The S9.6 staining protocol used was as previously described[14]. Coverslips were treated with RNase A (Fermentas) for 1 h at 37 °C and where indicated with RNase H (NEB) for 24 h at 37 °C prior to blocking. Coverslips were then incubated with 1:100 diluted S9.6 antibody (Kerafast) for 36 h at 4 °C. Cells were mounted and the entire nucleus imaged using LSM780 or LSM710 confocal microscope (Carl Zeiss, Germany), using a ×63/1.40 Oil DIC M27 Plan-ApoChromat objective. Detection of R-loops using RNase H1$^{D210N}$-V5 was carried out as previously described[68]. In brief, cells transfected with RNase H1$^{D210N}$-V5 were pre-extracted in ice-cold PBS/0.2% Triton-X 100 for 2 min, fixed in 4% (w/v) paraformaldehyde, permeabilised in 0.1% Triton-X 100, blocked with 1% (w/v) BSA in 1% (w/v) FBS in PBS, incubated with 1:500 diluted V5 antibody (ThermoFisher Scientific) for 1 h at room temperature before confocal imaging. DAPI was excited with laser line 405 nm and emission collected between 410 and 495 nm. Alexa-fluor 488 was excited with laser line 488 nm and emission collected between 495 and 596 nm. Alexa-fluor 594 was excited with laser line 594 nm and emission collected between 599 and 734 nm. Alexa-fluor 647 was excited with laser line 633 nm and emission collected between 638 and 755 nm. The 5′EU, S9.6 and V5 mean nuclear intensity signal (including the nucleolus) was determined using the ImageJ plugin/algorithm kindly provided by Dr Kienan Savage, Queen's University Belfast[69]. A minimum of 100 cells was included per treatment.

**RNA-seq**. RNA was extracted using TRI reagent (Sigma) and treated with DNase I (NEB). The samples were purified using the RNeasy MinElute Clean-up kit (Qiagen) and purity confirmed using a Bioanalyzer. The mRNA fraction was selected for reverse transcription to cDNA. The cDNA generated was end-repaired, A-tailed and adaptor-ligated. cDNA libraries were size selected, multiplexed, tested for quality and subjected to paired-end sequencing in one lane of a flow cell. Accession for RNA-seq data is GSE157371.

**RNA-seq differential expression analysis**. The analyses were run through an established RNA-seq pipeline by the Buffa lab. The technical adaptors in the pair-end raw reads were trimmed out by trimmomatic-0.32 and the clear reads were aligned to human genome version GRCh38 release 82 using tophat v2.0.13. The mapped reads from each gene were counted and normalised to fragments per kilobase of transcript per million (FPKM) using Cuffdiff-2.2.1. Fragments from PCR duplication were removed by Picard 1.124. The significance of fold changes of genes among samples was estimated by the nonparametric method Rank Product 3.3, a package in R[70–75].

**Correlation of SETX expression with TCGA RNA-seq data**. RNA-seq data (RNA Seq V2 RSEM) for 382 colorectal adenocarcinoma tumours, 501 lung squamous cell carcinomas and 517 lung adenocarcinomas were downloaded from the TCGA project (accessed through cBioportal: http://www.cbioportal.org/ on the 12th April

2017). To examine *SETX* expression against hypoxia, a validated hypoxia metagene signature was used[44]. To examine *SETX* expression against tumour-associated, hypoxia-induced p53 activity (referred to as hypoxic p53 targets in the figure), raw data for each sequenced gene were rescaled to set the median equal to 1, and hypoxic p53 targets were determined by quantifying the median expression of 6 p53 target genes associated with hypoxia-induced p53 activity (encoding *BTG2, CYFIP2, INPP5D, KANK3, PHLDA3,* and *SULF2*)[25].

**Prediction of RLFS sequences**. The quantitative model of R-loop forming sequences (qm-RLFS) was used to predict RLFS from fasta files of the human genome build GRCh38.p13 (accessed from https://www.gencodegenes.org/human/ on 23rd March 2020) using default settings. Bedtools[76] was used to merge RLFS with at least 1 base pair overlap and these merged regions were used for subsequent analysis.

Gene coordinates were accessed via BiomaRt using the Ensembl hg38 build version 100[77] and tested for overlaps with the coordinates of genes in the up and downregulated groups. Two kb flanks were added at the transcription start transcription termination sites to include promoter and terminator regions respectively. Only co-transcriptional RLFS were assessed. Custom R scripts were then used to quantify and plot RLFS per kilobase of gene length and per cent of a gene covered by RLFS. For senataxin knockdown vs. genome comparisons, each set of genes (up or downregulated by SETX KD vs. all protein-coding genes) was sampled 100,000 times, drawing 20% of the number of genes in the SETX KD group with each iteration. The mean of each RLFS metric was calculated for each iteration and the values plotted as probability densities and compared using two-tailed Student's $t$-tests.

**DNA fibre analysis**. For normoxic (21% $O_2$) treatments, cells were pulse-labelled with CldU (25 mM) for 20 min, washed once with fresh media, followed by the second label IdU (250 mM) for 20 min. For hypoxic (<0.1% $O_2$) treatments, cells were incubated in hypoxia for 3 h before being pulse-labelled with CldU (25 mM) for 2 h, washed once with fresh media, then followed by IdU (250 mM) for 1 h[78]. Fibres were spread, stained and mounted (Invitrogen) and imaged using LSM780 confocal microscope (Carl Zeiss Microscopy Ltd) using a ×63/1.40 Oil DIC M27 Plan-ApoChromat objective. Alexa-fluor 488 was excited with laser line 488 nm and emission collected between 493 and 551 nm. Alexa-fluor 555 was excited with laser line 543 nm and emission collected between 559 and 659 nm. Replication rates were calculated as $V_{IdU}$ (kb/min) = $[(x * 0.132 \text{ μm}) * 2.59 \text{ kb/μm}]/t$ (min), where $x$ = length of IdU.

**Statistical analysis**. Statistical tests were performed using GraphPad Prism software (GraphPad Software Inc.) and include one-way ANOVA, two-way ANOVA, two-tailed Student's $t$-test and Mann–Whitney U tests. ns = non-significant, *$p \leq$ 0.05, **$p \leq 0.01$, ***$p \leq 0.001$, ****$p \leq 0.0001$.

**Reporting summary**. Further information on research design is available in the Nature Research Reporting Summary linked to this article.

## Data availability
The sequencing data that support the findings reported in this study have been deposited in the National Center for Biotechnology Information Gene Expression Omnibus (GEO)

and are accessible through the GEO Series accession number GSE157371. All other relevant data are available from the corresponding author upon request. Source data are provided with this paper.

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

## Acknowledgements

We thank Monica Olcina and Kienan Savage for their helpful comments on the manuscript. We would like to acknowledge the skills and expertise of the staff in the Microscopy Scientific Research Facility at the Department of Oncology, University of Oxford. SR, KBL, PV and MH were supported by a CRUK grant C5255/A23755 (awarded to E.M.H.). N.N. was supported by an MRC studentship (MC_ST_U16007). I. P.F. was supported by CRUK Oxford Centre Prize DPhil Studentship C38302/A12981. N.G. was supported by a Royal Society University Research fellowship. W.-C.C. was funded by CRUK grant 23969 (awarded to F.M.B.). S.F.E.-K. was supported by a Wellcome Trust Investigator Award (103844) and a Lister Institute of Preventative Medicine Fellowship (137661). J.G. was supported by a Jean Shanks Foundation/ Pathological Society of Great Britain and Ireland Clinical PhD Fellowship (JSPS CPhD 2018 01).

## Author contributions

S.R. and T.M. designed, conducted, and interpreted the experiments and wrote the paper, N.N., I.P.F., and M.-S.H. conducted experiments, J.G., P.V., W.-C.C., and F.M.B. carried out the RNA-seq bioinformatics, K.B.L. performed the TCGA analysis and advised on experiments, N.G. and S.F.E.-K. advised on the project, E.M.H. conceived the project, designed/interpreted experiments and wrote the paper. All authors commented on the manuscript.

## Competing interests

The authors declare no competing interests.
