## [Peer Review File · Nature Communications]

Reviewers' comments:

Reviewer #1 (Remarks to the Author):

The manuscript by Ramachandran et al. uncovers a new role for SETX in the hypoxia response. They describe several novel findings including i) an increase in R-loop formation during hypoxia, ii) the involvement of SETX in the resolution of these R-loops during hypoxia, iii) the regulation of SETX and R-loop resolution by the PERK/ATF4 arm of the UPR during hypoxia, and iv) a requirement of SETX to prevent apoptosis and promote clonogenic cell survival. These are exciting, novel, and important observations. My comments and concerns are focused primarily on the consistency of the data presented across the figures and on the use and interpretation of the assays used.

Comments:

1) Figure 1 and description in text. SETX is described to correlate with a signature of p53-target genes specifically induced at <0.1% O₂. This is interpreted as correlating with hypoxia at these concentrations. However, presumably many of the TCGA tumors analyzed have mutations in p53? This needs to be accounted for in the analysis of this correlation.

2) Measurement of R-loops uses S9.6 nuclear intensity. While interesting, orthogonal assays to demonstrate the presence of R-loops should be considered.

3) Given the importance of identifying SETX as a key regulator of damage under hypoxia, it is important to confirm this interpretation through other means. Do you see assembly of other DNA repair proteins at these foci like you would for other types of DNA damage? What is the nature of the damage – can you see formation of double strand breaks or single strand breaks by COMET assay?

4) The apoptotic response is often dissociated from the DNA damage response and highly cell type dependent. Is the increase in apoptosis and decrease in clonogenic survival seen across all the cell lines examined?

5) It would be of interest to understand which downstream DDR pathways are critical for the type of damage induced upon loss of SETX. The data presented in figure 3 suggest that this damage is coupled to transcription and that R-loops may be involved. This would be rather easy to probe using various DNA repair chemical inhibitors to probe for involvement of double strand breaks.

6) In Figure 5A, hypoxia causes only a modest increase in R-loops. This is in contrast to the same experiment demonstrating a much larger increase in Figure 2A. This variability is unexplained. Furthermore, inhibition PERK has no impact on the cell line shown in Figure 5A (perhaps a small reduction in R-loops), whereas knockdown of PERK shown in Figure S8 causes a large increase R-loop formation. In this case the increase from PERK knockdown is larger than hypoxia. This is not explained and raises concerns over the interpretation of the S9.6 nuclear intensity.

7) On page 9 it is stated “PERK depletion or inhibition also partially rescued the global transcriptional repression observed in hypoxia (Figure 5B, S8C)”. This is not an accurate description of the data. Figure S8C shows a massive increase in 5'EU nuclear intensity. What is the basis for this? It seems highly unlikely this is due to increased levels of transcription. It creates concerns on the use and interpretation of the 5'EU assay as a measure of transcription. Orthogonal measures should be used to confirm the transcriptional changes, and the discrepancies between the data in Figure 5 and Figure S8 explained.

8) The importance of the UPR in signaling to SETX to protect cells during hypoxia should be confirmed using clonogenic and apoptotic assays.

9) It is interesting that reduction of SETX has a preferential effect on the transcription of rRNA processing, nucleolus and ribosome biogenesis. This may suggest a possible connection to UPR regulation of SETX given their involvement in translation that could be discussed in the Discussion.

Reviewer #2 (Remarks to the Author):

Ramachandran and colleagues report an increase in senataxin expression in hypoxia and linked it mechanistically to unfolded protein response. This manuscript addresses several important topics from regulation of senataxin to hypoxia response. However, my enthusiasm is relatively low. I find the manuscript difficult to follow. I am not convinced of either of the key findings: the increased in senataxin expression in hypoxia or that UPR regulates senataxin expression.

1. Senataxin is a very large gene so measurement of its expression levels by RT-PCR needs to include several sets of primers. I looked at the primers in Table S1. There is only one primer set per gene. The primers for several genes, including SETX, do not map to the human genome sequences by in silico PCR. Using the in silico PCR tool at the UCSC Genome Browser and several of the genome builds, I found primers for AQR, RNaseH1, TCF19, GHITM that mapped to those genes; but the ones for SETX, RNaseH2B, PIF1 did not map anywhere. Thus, I do not know which part of SETX was assayed for its expression levels. This is important since the manuscript is about the induction of senataxin expression in hypoxia. The readers must be convinced that this finding is made carefully.

2. I am not convinced by the model in figure 5H and the conclusion that PERK/ATF4 regulate senataxin expression. ER stress/ UPR shuts down ribosomal RNA synthesis to decrease protein synthesis. In most human cells, as much as 50% of R-loops are found in rRNA. With a decrease of rRNA in ER stress, R-loops also decrease. When cells are exposed to low oxygen, tunicamycin or thapsigargin that activates PERK / ATF4 pathway, there are fewer rRNA and subsequently fewer R-loops. The findings can be explained by formation of fewer R-loops or senataxin resolving more R-loops. The authors should tease apart these two pathways.

3. There are different isoforms of senataxin – are they all induced by hypoxia? The western blot data in Figure 1F-H show only a very small part of the gels so it is not possible for readers to see the different isoforms.

Reviewer #3 (Remarks to the Author):

In this manuscript Ramachandran et al., report the original observation of induction of the RNA helicase Senataxin under hypoxic conditions, specifically at less than 0.1% oxygen, which is considered as radiobiological hypoxia. Senataxin function in transcription termination and also contributes to the resolution of toxic DNA/RNA hybrids (also known as R-loops) that form upon interference between transcription and DNA replication. Authors show that in hypoxic conditions, Senataxin expression is upregulated at the transcriptional level, resulting in a concomitant increase in Senataxin protein abundance. The authors have also identified the factors involved in induction of Senataxin gene expression corresponding to the PERK/ATF4 pathway of the unfolded protein response.

Overall, the work is well executed and the data are convincing. Nevertheless, this remains an original observation that requires further investigations to unveil its truly functional significance. Is there a specific importance in having Senataxin upregulated in hypoxia conditions? All the phenotypes reported upon Senataxin downregulation in hypoxic conditions have already been reported in normoxia conditions. Authors claim that these phenotypes are due to its R-loop-resolving activity, but Senataxin has other cellular roles, such as transcription of coding and non-coding RNAs, as well as in transcriptional coupled DNA repair (TCR). Authors have not really investigated this point. Can the authors design experiments to distinguish between these functions? Further, the observation that hypoxia represses transcription is at odd with the both the reported data and the claims made by the authors related to the occurrence of R-loops as conflicts between transcription and replication.

Specific points

1. Page 6, second paragraph, line 4, authors write that in hypoxia transcription is repressed, however, few lines underneath, authors claim that SETX depletion in hypoxia caused 341 genes to

increase in expression, and further that “to test whether the DNA damage observed in the SETX depleted hypoxic cells was linked to transcription”, they treated cells with dichloro-beta-D-ribofuranosylbenzimidazole (DRB), which inhibits Pol II-mediated transcription. Authors report that Senataxin silencing in hypoxia affects the expression over 300 genes: how does this work? Is this a consequence of a general function of Senataxin in transcription or there is a specific function at this genes subset? How many genes are globally induced in hypoxia? The authors must clarify this point by stating that hypoxia strongly reduced but does not completely repress transcription. In addition, if transcription is repressed, and if R-loops are a result of conflicts between transcription and replication, why R-loops increase? All these statements are contradictory.

2. Authors report that some other factors involved in R-loops processing are not upregulated in hypoxia, but this does not mean that they do not function. It would then be a good control to show that silencing of any of these genes would not (or would?) generate phenotypes similar to Senataxin silencing in hypoxia. Authors also report that Senataxin silencing induced DNA damage in hypoxia. In normoxia, DNA damage generated by R-loops is processed through the NER, which generated DNA double strand breaks. However, the authors claim that NER is silent in hypoxia. Hence, is NER reactivated upon Senataxin silencing? Otherwise, how this DNA damage is generated? Is this related to the Senataxin function in TCR? Further, authors claim that they have demonstrated increased Senataxin levels in radiobiological hypoxia conditions; however, no specific experiments involving irradiation have been presented. Does Senataxin silencing make cells more sensitive to radiation under hypoxic conditions? Hence, their observations are limited to cells exposed to low oxygen concentration and correlated to publically available datasets.

3. Another point is related to the cell cycle. It is known that under hypoxia cells arrest at the G1/S transition. Hence, what is the fraction of cells in S-phase and is this changing upon Senataxin downregulation? A FACS analysis of these cells must be shown with BrdU incorporation rates. In other words, the effect seen upon Senataxin downregulation on replication fork progression is due to an effect on cell cycle, transcription or R-loops? Authors have not challenged this experiment with RNaseH1 (Figure 2D-F).

Other minor points:

In the introduction, authors refer to several enzymes involved in R-loop resolution by citing two outdated reviews. These must be replaced by references that are more recent. Since 2014, other new enzymes involved in R-loop resolution have been described.

Authors use 53BP1 as a marker of DNA damage. They should explain that 53BP1 is a marker of DNA double strand breaks.

The weak reduction in EU intensity of Figure S3A lack statistical significance. Statistical significance values are missing in Figure S5F-G

re NCOMMS-20-19058

Reviewer #1 (Remarks to the Author):

The manuscript by Ramachandran et al. uncovers a new role for SETX in the hypoxia response. They describe **several novel findings** including i) an increase in R-loop formation during hypoxia, ii) the involvement of SETX in the resolution of these R-loops during hypoxia, iii) the regulation of SETX and R-loop resolution by the PERK/ATF4 arm of the UPR during hypoxia, and iv) a requirement of SETX to prevent apoptosis and promote clonogenic cell survival. **These are exciting, novel, and important observations.** My comments and concerns are focused primarily on the consistency of the data presented across the figures and on the use and interpretation of the assays used.

Thank you for your comments and in particular for highlighting the novelty of our work.

Comments:

1) Figure 1 and description in text. SETX is described to correlate with a signature of p53-target genes specifically induced at <0.1% O₂. This is interpreted as correlating with hypoxia at these concentrations. However, presumably many of the TCGA tumors analyzed have mutations in p53? This needs to be accounted for in the analysis of this correlation.

The group of 6 p53-dependent hypoxia-inducible genes do indeed predict for the p53 status of the samples in the TCGA (demonstrated in Leszczynska et al, JCI 2015). We are therefore showing that in a subset of the TCGA (p53 wild type) there is evidence that SETX is induced in response to hypoxia (<0.1% O₂). As there is not a validated signature for these hypoxic conditions (<0.1% O₂), we feel this is the optimal approach to interrogate patient samples. The text has been modified to improve clarity.

2) Measurement of R-loops uses S9.6 nuclear intensity. While interesting, orthogonal assays to demonstrate the presence of R-loops should be considered.

We are well aware of the limitations of the S9.6 antibody and have employed both IF and slot blot assays as a result. We have also used expression of the V5 tagged RNase H^{D210N} catalytic-dead mutant to visualise R-loops by immunofluorescence (Chen et al. Mol Cell 2017, Nguyen et al. Mol Cell 2017, Klusmann et al. PNAS 2018). This data is although preliminary supports the conclusion that R-loops accumulate in hypoxia. CPT was used as a positive control and DRB as a negative control (Figure 1). Once the lab is fully reopened and specifically the microscopy facility, we would be happy to repeat this experiment so that it could be included in a revised manuscript.

Figure 1. R-loops detected by trapped RNase H^{D210N} accumulate in response to Hypoxia and CPT but not DRB. A. A549 cells were transfected with a plasmid expressing V5-tagged RNase H^{D210N} followed by exposure to 21% O₂ (Norm), <0.1% O₂ (Hyp), DRB (100 μM) or CPT () for 6 hours. Cells were then fixed and stained with a V5 antibody and DAPI to visualise the nucleus. Representative images are shown. B. Quantification of stained cells are shown. (n=1)

3) Given the importance of identifying SETX as a key regulator of damage under hypoxia, it is important to confirm this interpretation through other means. Do you see assembly of other DNA repair proteins at these foci like you would for other types of DNA damage? What is the nature of the damage – can you see formation of double strand breaks or single strand breaks by COMET assay?

We have determined that 53BP1 is the best assay to use to assess DNA damage in hypoxic conditions (Olcina et al., Mol Cell 2013). Obvious alternatives include BRCA1 and γH2AX. However, BRCA1 is repressed in hypoxic conditions (Chan et al., Can Res 2010), and γH2AX forms pan-nuclear staining making it difficult to spot the occurrence of foci (Bencokova et al., MCB 2009). We have not carried out Comet assays in this instance, although have used this assay previously to show damage does not accumulate in hypoxia alone (Hammond et al., MCB 2002). Figure 1 has been updated to strengthen and support this point.

4) The apoptotic response is often dissociated from the DNA damage response and highly cell type dependent. Is the increase in apoptosis and decrease in clonogenic survival seen across all the cell lines examined?

We agree that these responses are often un-coupled. We have not carried out these experiments in more than the A549 cell line but could do if required.

5) It would be of interest to understand which downstream DDR pathways are critical for the type of damage induced upon loss of SETX. The data presented in figure 3 suggest that this damage is coupled to

transcription and that R-loops may be involved. This would be rather easy to probe using various DNA repair chemical inhibitors to probe for involvement of double strand breaks.

We have not used chemical inhibitors and caution is warranted as both inhibition of ATR and ATM lead to DNA damage in hypoxia (Pires et al., *Can Res* 2010).

6) In Figure 5A, hypoxia causes only a modest increase in R-loops. This is in contrast to the same experiment demonstrating a much larger increase in Figure 2A. This variability is unexplained. Furthermore, inhibition PERK has no impact on the cell line shown in Figure 5A (perhaps a small reduction in R-loops), whereas knockdown of PERK shown in Figure S8 causes a large increase R-loop formation. In this case the increase from PERK knockdown is larger than hypoxia. This is not explained and raises concerns over the interpretation of the S9.6 nuclear intensity.

Figure 5A and 2A cannot be directly compared as in 5A cells were exposed to hypoxia for 6 hours and in 2A they were exposed for 12 hours. This variability has now been explained in the text.

7) On page 9 it is stated "PERK depletion or inhibition also partially rescued the global transcriptional repression observed in hypoxia (Figure 5B, S8C)". This is not an accurate description of the data. Figure S8C shows a massive increase in 5'EU nuclear intensity. What is the basis for this? It seems highly unlikely this is due to increased levels of transcription. It creates concerns on the use and interpretation of the 5'EU assay as a measure of transcription. Orthogonal measures should be used to confirm the transcriptional changes, and the discrepancies between the data in Figure 5 and Figure S8 explained.

We apologise for our previous inaccuracy. The revised version of the manuscript no longer includes the data demonstrating the impact of PERK on global transcription rates as we feel this was confusing and not central to the manuscripts main findings.

For the interest of the reviewer; It has been shown that agents which lead to ER stress/UPR also reduce rRNA synthesis through a PERK-dependent mechanism (DuRose et al., *MCB* 2009), and therefore it is plausible that PERK depletion in hypoxia would lead to a partial rescue of transcription rates. We have confirmed our findings using qPCR for the nascent 47S transcript which as the precursor to rRNA reflects the majority of cellular transcription. We have decided to prepare a separate manuscript focused on hypoxia-mediated repression of RNA polymerase I activity and will include these data.

8) The importance of the UPR in signaling to SETX to protect cells during hypoxia should be confirmed using clonogenic and apoptotic assays.

Decreased colony survival with PERK loss in hypoxia has been shown previously (Rouschop et al, *PNAS USA* 2013, Koumenis et al, *MCB* 2002). It has also been shown that PERK loss leads to increased apoptosis in hypoxia (Bi et al, *EMBO J* 2005).

We have preliminary data, which indicates that PERK inhibition leads to decreased colony survival in hypoxia, we could expand on this if the reviewers would like this included (figure 2).

Figure 2. Colony survival assay of A549 cells treated with DMSO or PERK inhibitor GSK2606414 (10 μ M) exposed to <0.1% O₂ for 0, 6, 18, 24, 42, 48 hours. The indicated stats represent a 2-way ANOVA test representing variability due to the PERK inhibitor. Errors bars show S.D. from technical replicates from 1 biological experiment.

9) It is interesting that reduction of SETX has a preferential effect on the transcription of rRNA processing, nucleolus and ribosome biogenesis. This may suggest a possible connection to UPR regulation of SETX given their involvement in translation that could be discussed in the Discussion.

Indeed, an interesting point and given the recent publication (Abraham et al., Nature 2020) one that we are very interested to follow up. However, we believe this is too complex to include here and would result in a lack of focus for this manuscript.

Reviewer #2 (Remarks to the Author):

Ramachandran and colleagues report an increase in senataxin expression in hypoxia and linked it mechanistically to unfolded protein response. This manuscript **addresses several important topics** from regulation of senataxin to hypoxia response. However, my enthusiasm is relatively low. I find the manuscript difficult to follow. I am not convinced of either of the key findings: the increased in senataxin expression in hypoxia or that UPR regulates senataxin expression.

We appreciate that our manuscript was over-complicated and have endeavoured to simply and focus on the main findings in the revised version.

1. Senataxin is a very large gene so measurement of its expression levels by RT-PCR needs to include several sets of primers. I looked at the primers in Table S1. There is only one primer set per gene. The primers for several genes, including SETX, do not map to the human genome sequences by in silico PCR. Using the in-silico PCR tool at the UCSC Genome Browser and several of the genome builds, I found primers for AQR, RNaseH1, TCF19, GHITM that mapped to those genes; but the ones for SETX, RNaseH2B, PIF1 did

not map anywhere. Thus, I do not know which part of SETX was assayed for its expression levels. This is important since the manuscript is about the induction of senataxin expression in hypoxia. The readers must be convinced that this finding is made carefully.

All primers used for RT-qPCR were checked using the NCBI Primer Blast tool against the Homo Sapiens Refseq mRNA database for primer specificity, and in each case the qPCR denaturation curve was checked to ensure that a single distinct peak was observed to indicate a single species being detected. Most of the primers were designed to span an intron in order to specifically amplify the cDNA and not the genomic DNA. The SETX primers map to exon 25 -26, and this was verified in the hg38 human genome. The SETX primers used have also been published previously (Skourti-Stathaki et al. Molecular Cell 2011). In addition, we have shown that 2 different SETX siRNA led to a decrease in transcript as tested by these primers. Finally, our RNA-seq data confirmed that SETX is induced 3-fold in 3 independent biological repeats.

2. I am not convinced by the model in figure 5H and the conclusion that PERK/ATF4 regulate senataxin expression. ER stress/ UPR shuts down ribosomal RNA synthesis to decrease protein synthesis. In most human cells, as much as 50% of R-loops are found in rRNA. With a decrease of rRNA in ER stress, R-loops also decrease. When cells are exposed to low oxygen, tunicamycin or thapsigargin that activates PERK / ATF4 pathway, there are fewer rRNA and subsequently fewer R-loops. The findings can be explained by formation of fewer R-loops or senataxin resolving more R-loops. The authors should tease apart these two pathways.

In support of our conclusion that PERK/ATF4 regulates SETX;

- Inhibition of PERK leads to reduced SETX in hypoxia
- Depletion of PERK leads to reduced SETX in hypoxia
- Depletion of ATF4 leads to reduced SETX in hypoxia
- ATF4 accumulates at a region of the SETX promoter which also contains an ATF4 binding site (ChIP)
- SETX is also induced by tunicamycin and thapsigargin, both of which induce a UPR

The author suggests a model for the biological response to hypoxia which includes a decrease in R-loops, however, our data demonstrates that despite the induction of a UPR in hypoxia, R-loops accumulate. This finding is now also supported by a recent paper which also demonstrates an increase in R-loops in response to induction of a UPR (using pharmacological agents as opposed to hypoxia) (Choo et al., Cell Death Dis 2020).

3. There are different isoforms of senataxin – are they all induced by hypoxia? The western blot data in Figure 1F-H show only a very small part of the gels so it is not possible for readers to see the different isoforms.

According to NCBI, there are 3 major transcripts for human SETX. Transcript 1 and 2 differ in the 5'UTR and produce a protein of 2677 amino acids (predicted size approx. 303 kDa), while an alternative in-frame exon in transcript 3 leads to a slightly larger protein of 2706 amino acids expected to be around 306 kDa. The

small size difference between the two isoforms cannot be detected by western blot. In publications including SETX westerns blots, one band for SETX is shown. The primers we have used to detect SETX do not distinguish between isoforms i.e. they detect all 3, therefore we are unable to determine if the hypoxia-mediated induction of SETX is isoform specific.

Reviewer #3 (Remarks to the Author):

In this manuscript Ramachandran et al., report the **original observation** of induction of the RNA helicase Senataxin under hypoxic conditions, specifically at less than 0.1% oxygen, which is considered as radiobiological hypoxia. Senataxin function in transcription termination and also contributes to the resolution of toxic DNA/RNA hybrids (also known as R-loops) that form upon interference between transcription and DNA replication. Authors show that in hypoxic conditions, Senataxin expression is upregulated at the transcriptional level, resulting in a concomitant increase in Senataxin protein abundance. The authors have also identified the factors involved in induction of Senataxin gene expression corresponding to the PERK/ATF4 pathway of the unfolded protein response.

Overall, the work is well executed and the data are convincing. Nevertheless, this remains an original observation that requires further investigations to unveil its truly functional significance. Is there a specific importance in having Senataxin upregulated in hypoxia conditions? All the phenotypes reported upon Senataxin downregulation in hypoxic conditions have already been reported in normoxia conditions. Authors claim that these phenotypes are due to its R-loop-resolving activity, but Senataxin has other cellular roles, such as transcription of coding and non-coding RNAs, as well as in transcriptional coupled DNA repair (TCR). Authors have not really investigated this point. Can the authors design experiments to distinguish between these functions? Further, the observation that hypoxia represses transcription is at odd with the both the reported data and the claims made by the authors related to the occurrence of R-loops as conflicts between transcription and replication.

Thank you for your positive comments. Our apologies for not being clear on the effect of hypoxia on transcription. While there is a significant global reduction of transcription in hypoxia it does not lead to a complete repression of transcription. For example, certain transcription factors including HIF, p53 and ATF4 are highly active in hypoxia.

We have investigated the role of Senataxin in the transcription of coding RNAs by RNA-sequencing, which showed that SETX had a more significant effect in hypoxia as compared to normoxia, highlighting the importance of SETX to the hypoxia response.

Specific points

1. Page 6, second paragraph, line 4, authors write that in hypoxia transcription is repressed, however, few lines underneath, authors claim that SETX depletion in hypoxia caused 341 genes to increase in expression, and further that “to test whether the DNA damage observed in the SETX depleted hypoxic cells was linked to transcription”, they treated cells with dichloro-beta-D-ribofuranosylbenzimidazole (DRB), which inhibits

Pol II-mediated transcription. Authors report that Senataxin silencing in hypoxia affects the expression over 300 genes: how does this work? Is this a consequence of a general function of Senataxin in transcription or there is a specific function at this genes subset? How many genes are globally induced in hypoxia? The authors must clarify this point by stating that hypoxia strongly reduced but does not completely repress transcription. In addition, if transcription is repressed, and if R-loops are a result of conflicts between transcription and replication,

why R-loops increase? All these statements are contradictory.

We clearly failed to describe these data adequately and we apologies for that. This has now been corrected. Regarding discrepancies between transcription and R-loops, this is not that surprising since in many trinucleotide expansion diseases (the focus of the Gromak lab), there is a decreased transcription on expanded genes but specific accumulation of R-loops at these loci. Similarly, treating cells with the Topoisomerase 1 inhibitor, CPT, leads to transcriptional defects but some specific R-loops accumulate. Currently, there is not a clear indication from the literature that there is a direct correlation between transcription and R-loops, however transcription is certainly needed for R-loop formation.

2. Authors report that some other factors involved in R-loops processing are not upregulated in hypoxia, but this does not mean that they do not function. It would then be a good control to show that silencing of any of these genes would not (or would?) generate phenotypes similar to Senataxin silencing in hypoxia. Authors also report that Senataxin silencing induced DNA damage in hypoxia. In normoxia, DNA damage generated by R-loops is processed through the NER, which generated DNA double strand breaks. However, the authors claim that NER is silent in hypoxia. Hence, is NER reactivated upon Senataxin silencing? Otherwise, how this DNA damage is generated? Is this related to the Senataxin function in TCR? Further, authors claim that they have demonstrated increased Senataxin levels in radiobiological hypoxia conditions; however, no specific experiments involving irradiation have been presented. Does Senataxin silencing make cells more sensitive to radiation under hypoxic

conditions? Hence, their observations are limited to cells exposed to low oxygen concentration and correlated to publically available datasets.

The reviewer has a good point in that while some other R-loop processing factors are repressed in hypoxia, residual protein could be functional. We will amend the discussion to reflect this but cannot commit to fishing for other genes that play a role at this stage.

Apologies, we did not intend to give the impression that NER is completely silent in hypoxia. While DNA repair including NER is repressed in hypoxia the repression is not instant and requires longer time points than 6 hours (which is the time point at which DNA damage was observed with SETX knock-down in hypoxia). Therefore, it is plausible that NER could still function to lead to the DNA breaks observed. Alternatively, other functions of SETX including in TCR could be leading to the DNA breaks observed upon SETX depletion in hypoxia. For clarity and because of space constraints this section has been removed.

Our use of the term 'radiobiological hypoxia' was mis-leading as this paper is not radiation-focused (as much of the work in the Hammond lab is) and so has been removed.

3. Another point is related to the cell cycle. It is known that under hypoxia cells arrest at the G1/S transition. Hence, what is the fraction of cells in S-phase and is this changing upon Senataxin downregulation? A FACS analysis of these cells must be shown with BrdU incorporation rates. In other words, the effect seen upon Senataxin downregulation on replication fork progression is due to an effect on cell cycle, transcription or R-loops? Authors have not challenged this experiment with RNaseH1 (Figure 2D-F).

We have carried out the suggested experiments. As expected exposure to hypoxia led to a significant drop in BrdU incorporating cells (compare S phase (grey bar) between A and B). Cell cycle (BrdU/PI) analysis by FACS demonstrated that SETX knock-down lead to a decrease in the level of BrdU incorporating (S phase) cells in normoxia, but in hypoxia where there is already a significant decrease in the level of BrdU incorporating (S phase) cells there was no further significant decrease with SETX depletion (Figure 3). These data are what led us to use the more sensitive DNA fibre assay to determine the SETX-dependent changes to DNA replication in hypoxia (Figure 2F of manuscript). The data shown below are however in agreement with the DNA fibre data and demonstrate that there is not a significant SETX-dependent effect on cell cycle distribution in hypoxia.

Figure 3. HCT116 cells were treated with mock and SETX siRNA and exposed to **A.** 21% O₂ (Norm) or **B.** Hypoxia (<0.1% O₂) for 6 hours. In the last hour of treatment, cells were exposed to BrdU (20 mM), fixed and stained with BrdU/PI and analysed by flow cytometry. G1 and G2 were determined by DNA content while the cells in S-phase were determined by positive BrdU staining. n=3

We would be happy to include these data (Figure 3) in a revised manuscript although please note the HCT116 cell line was used as opposed to the A549s used in the majority of figures.

We have attempted the experiment suggested by the reviewer using RNase H1 expression to rescue replication rates in hypoxia. Although preliminary, these data are encouraging (Figure 4). A partial rescue of replication rates with RNase H1 overexpression in hypoxia was observed, supporting the conclusion that R loops contribute to replication stress in hypoxia. We have not pursued these experiments (due to lockdown) but also as we would not expect a complete rescue as since limiting nucleotides is a key contributor to replication stress in hypoxia (Foskolou et al, Mol Cell 2017). Therefore, we believe this approach will generate data which will be difficult to interpret.

[REDACTED]

Other minor points:

In the introduction, authors refer to several enzymes involved in R-loop resolution by citing two outdated reviews. These must be replaced by references that are more recent. Since 2014, other new enzymes involved in R-loop resolution have been described.

References have been changed.

Authors use 53BP1 as a marker of DNA damage. They should explain that 53BP1 is a marker of DNA double strand breaks.

For the reasons already given, 53BP1 is the optimal DNA damage marker in hypoxia.

The weak reduction in EU intensity of Figure S3A lack statistical significance. Statistical significance values are missing in Figure S5F-G

During the revision process and with a view to conveying our principle findings more clearly some data has not been repeated as it is no longer part of the manuscript (Figure S3A).

Due to lack of access to both the main lab and the microscopy facility we have been unable to repeat Figures S5F-G to determine the statistical significance. These experiments can be repeated in time for a revised manuscript.

REVIEWER COMMENTS

Reviewer #1 (Remarks to the Author):

The authors have addressed the majority of initial criticisms previously raised. However, due in part to COVID related restrictions, some of the additional data are incomplete. The following areas should be completed. Below are original numbered criticisms that require further attention.

2) Measurement of R-loops uses S9.6 nuclear intensity. While interesting, orthogonal assays to demonstrate the presence of R-loops should be considered.

- the additional assay using V5 tagged RNase HD210N catalytic-dead mutants should be completed and included

6) In Figure 5A, hypoxia causes only a modest increase in R-loops. This is in contrast to the same experiment demonstrating a much larger increase in Figure 2A. This variability is unexplained. Furthermore, inhibition PERK has no impact on the cell line shown in Figure 5A (perhaps a small reduction in R-loops), whereas knockdown of PERK shown in Figure S8 causes a large increase R-loop formation. In this case the increase from PERK knockdown is larger than hypoxia. This is not explained and raises concerns over the interpretation of the S9.6 nuclear intensity.

- The authors state that "Figure 5A and 2A cannot be directly compared as in 5A cells were exposed to hypoxia for 6 hours and in 2A they were exposed for 12 hours." That may be true but still requires demonstration of confidence in the assay. The experiment should be repeated in a comparable way. The inconsistency with PERK knockdown and PERK inhibition also needs to be examined and explained.

7) On page 9 it is stated "PERK depletion or inhibition also partially rescued the global transcriptional repression observed in hypoxia (Figure 5B, S8C)". This is not an accurate description of the data. Figure S8C shows a massive increase in 5'EU nuclear intensity. What is the basis for this? It seems highly unlikely this is due to increased levels of transcription. It creates concerns on the use and interpretation of the 5'EU assay as a measure of transcription. Orthogonal measures should be used to confirm the transcriptional changes, and the discrepancies between the data in Figure 5 and Figure S8 explained.

The authors responded "We apologise for our previous inaccuracy. The revised version of the manuscript no longer includes the data demonstrating the impact of PERK on global transcription rates as we feel this was confusing and not central to the manuscripts main findings." Removing this data does remove confusion, but it does not explain the unexpected finding. My issue with the data lies in the confidence of the assay and an additional approach or additional controls to validate the assay are needed given this previous highly unexplained result.

8) The importance of the UPR in signaling to SETX to protect cells during hypoxia should be confirmed using clonogenic and apoptotic assays.

The authors responded by including a preliminary figure testing the consequences of a PERK inhibitor on hypoxia tolerance. As indicated by the authors, this has previously been demonstrated by others. My original request was to demonstrate the requirement of SETX in this effect. Clonogenic assays with SETX manipulation would establish this and should be included.

Reviewer #2 (Remarks to the Author):

The revision answers some of my concerns but not all. I understand that it is hard to carry out experiments at a time of COVID shutdown. Whether this revision meets the journal standard under these special circumstances is an editorial decision.

Reviewer #3 (Remarks to the Author):

I consider that this work is still too preliminary for publication. Although the authors have provided few new data, these remain insufficient. This work has a potential but needs more work. The authors failed to provide a clear explanation of what is the role, if any, of Senataxin induction in hypoxia. Authors state in the abstract that "Senatxin plays a key role in protecting cells from DNA damage induced during transcription in hypoxia", however the molecular basis of this remains unclear.

re NCOMMS-20-19058A-Z

Reviewer #1 (Remarks to the Author):

The authors have addressed the majority of initial criticisms previously raised. However, due in part to COVID related restrictions, some of the additional data are incomplete. The following areas should be completed. Below are original numbered criticisms that require further attention.

Many thanks to the reviewer for acknowledging that we have previously addressed the majority of the criticisms raised. As suggested, we have focused on further addressing the specific list below.

2) Measurement of R-loops uses S9.6 nuclear intensity. While interesting, orthogonal assays to demonstrate the presence of R-loops should be considered.

- the additional assay using V5 tagged RNase HD210N catalytic-dead mutants should be completed and included

All R-loop experiments have now been repeated with RNase H^{D210N} mutant ($n=3$) and the data included in the manuscript. In addition, we provide quantification of the slot blots previously included. As a result, the R-loop data is supported by S9.6 IF, S9.6 slot blot and the use of RNase H^{D210N}.

6) In Figure 5A, hypoxia causes only a modest increase in R-loops. This is in contrast to the same experiment demonstrating a much larger increase in Figure 2A. This variability is unexplained. Furthermore, inhibition PERK has no impact on the cell line shown in Figure 5A (perhaps a small reduction in R-loops), whereas knockdown of PERK shown in Figure S8 causes a large increase R-loop formation. In this case the increase from PERK knockdown is larger than hypoxia. This is not explained and raises concerns over the interpretation of the S9.6 nuclear intensity.

- The authors state that "Figure 5A and 2A cannot be directly compared as in 5A cells were exposed to hypoxia for 6 hours and in 2A they were exposed for 12 hours." That may be true but still requires demonstration of confidence in the assay. The experiment should be repeated in a comparable way. The inconsistency with PERK knockdown and PERK inhibition also needs to be examined and explained

We apologise for the data inconsistencies observed in what was Fig 2A and 5A. The addition of the data generated using the RNase H^{D120N} method has resolved these issues. Importantly, the

RNase H^{D120N} approach further validated our previous conclusions i.e. that R-loops accumulate in hypoxia and that this can be further increased with loss/inhibition of SETX and PERK.

7) On page 9 it is stated "PERK depletion or inhibition also partially rescued the global transcriptional repression observed in hypoxia (Figure 5B, S8C)". This is not an accurate description of the data. Figure S8C shows a massive increase in 5'EU nuclear intensity. What is the basis for this? It seems highly unlikely this is due to increased levels of transcription. It creates concerns on the use and interpretation of the 5'EU assay as a measure of transcription. Orthogonal measures should be used to confirm the transcriptional changes, and the discrepancies between the data in Figure 5 and Figure S8 explained.

The authors responded "We apologise for our previous inaccuracy. The revised version of the manuscript no longer includes the data demonstrating the impact of PERK on global transcription rates as we feel this was confusing and not central to the manuscripts main findings." Removing this data does remove confusion, but it does not explain the unexpected finding. My issue with the data lies in the confidence of the assay and an additional approach or additional controls to validate the assay are needed given this previous highly unexplained result.

We completely agree with the reviewer that removing the data does not explain the discrepancy and furthermore, does not instil confidence of the EU assay. This was also of concern to us and therefore, we took great care to validate our EU assay. We have investigated the effect hypoxia on transcription in a separate manuscript from our group (Ng *et al.*, in preparation) which focusses on rDNA transcription in hypoxia. For your information we have included here data from three separate assays, EU incorporation, qPCR for the 47S pre-rRNA and northern blot for the 47S pre-rRNA, which all demonstrate that hypoxia (<0.1% O₂) and pharmacological transcriptional inhibitors (DRB, CX5461, low dose actinomycin D) reduce transcription (unpublished data). These data highlight that EU incorporation mirrors that of the qPCR and northern blot of the 47S pre-rRNA, therefore providing confidence that EU incorporation is a valid assay to determine global transcription, the cellular bulk of which is RNA polymerase 1 mediated.

[REDACTED]

In reference to our data shown in the first version of this manuscript suggesting that the PERK branch of the UPR represses rRNA, this is supported by a published study, which demonstrates that the PERK/EIF2 α arm of the UPR inactivates rDNA transcription upon pharmacological activation of the UPR (DuRose *et al.*, MCB 2009). As the UPR is active in hypoxic conditions (<0.1% O₂), it seems likely that UPR signalling in hypoxia may also inactivate rDNA transcription. This hypothesis is supported by the data shown in the original version of this manuscript i.e. that PERK inhibition increased transcription measured by EU incorporation. As this data has been previously reported (DuRose *et al.*, MCB 2009) and caused confusion in the first version of our manuscript we have chosen to not include it in the current version.

8) The importance of the UPR in signaling to SETX to protect cells during hypoxia should be confirmed using clonogenic and apoptotic assays.

The authors responded by including a preliminary figure testing the consequences of a PERK inhibitor on hypoxia tolerance. As indicated by the authors, this has previously been demonstrated by others. My original request was to demonstrate the requirement of SETX in this effect. Clonogenic assays with SETX manipulation would establish this and should be included. Thank you for clarifying this point, please note clonogenics plus and minus SETX siRNA were included in the original manuscript (Figure 3D, now 4E). The levels of apoptosis +/- SETX are also shown (Figure 4F and S4F).

Reviewer #2 (Remarks to the Author):

The revision answers some of my concerns but not all. I understand that it is hard to carry out experiments at a time of COVID shutdown. Whether this revision meets the journal standard under these special circumstances is an editorial decision.

Thank you for acknowledging the challenging times we all find ourselves in. It has indeed been difficult to carry out all of the experiments we would have liked to. However, we believe our revised manuscript is now greatly strengthened by the inclusion throughout of the orthogonal assay for R-loops (Figures 3C, 6C, S7C).

In addition, we have included new and supportive data. Specifically, we have further investigated the changes in gene expression identified in our +/- SETX RNA-seq. We used a validated R-loop prediction tool (quantitative model of R-loop forming sequences) and found that genes regulated by SETX under hypoxic conditions have a significantly higher density of R-loop forming sequences than protein coding genes selected at random. These data provide a clear link between hypoxia, SETX and R-loops and therefore has been included early in the manuscript (Figure 2D, E).

One of the most novel findings in our manuscript is the finding that R-loop levels accumulate in hypoxia (now shown by RNase H^{D210N} assay and using S9.6). We have expanded this to include analysis of the potential impact of these R-loops on the biological response to hypoxia. Now included are data which demonstrate that the hypoxia-induction of chromatin marks, previously linked to R-loops, is affected by RNase H1 over-expression (Figure 3E); hypoxia-induced replication stress is reduced by RNase H1 (Figure 3F, G) and finally, hypoxia-induced ATM activation is

reduced by RNase HI (Figure 3H). Together, these data fully support our conclusion that R-loops accumulate in hypoxia but, more importantly, also demonstrate that they are biologically significant.

Reviewer #3 (Remarks to the Author):

I consider that this work is still too preliminary for publication. Although the authors have provided few new data, these remain insufficient. This work has a potential but needs more work. The authors failed to provide a clear explanation of what is the role, if any, of Senataxin induction in hypoxia. Authors state in the abstract that "Senataxin plays a key role in protecting cells from DNA damage induced during transcription in hypoxia", however the molecular basis of this remains unclear.

We thank you the reviewer for acknowledging the potential of our study and also the interesting questions it raises. We believe that as the result of the addition of new data (listed for the previous reviewer), our study is no longer preliminary and provides novel insight, which will be of significant interest to the field. In addition to the list of new data above, we list here the genuinely novel messages from our manuscript and hope that the reviewer will agree that they are sufficient to warrant publication in Nature Communications:

- The physiologically relevant stress of hypoxia leads to the accumulation of R-loops despite decreased transcription and replication.
- SETX is induced in an oxygen dependent manner and by pharmacological inducers of the unfolded protein response (UPR). SETX has not previously been shown to be stress responsive.
- SETX expression is controlled by the PERK/ATF4 arm of the UPR in hypoxic conditions providing a novel link between R-loops and the UPR. We believe that it is very likely that others factors involved in R-loop resolution might be controlled by UPR-mediated signalling and that hypoxia is the ideal context to characterise these relationships.
- Inhibition of the UPR kinase, PERK leads to a further accumulation of R-loops in hypoxia.

REVIEWERS' COMMENTS

Reviewer #1 (Remarks to the Author):

The additional data has addressed all of my original criticisms and I do not have any other concerns.

Reviewer #2 (Remarks to the Author):

I appreciate the new data and the difficulties to carry out some experiments during COVID. However to link senataxin to hypoxia and ER stress - despite the new data, there are logical leaps that were made. At this point, it is up to the editors on whether this manuscript meet the minimum threshold for publication.

Reviewer #3 (Remarks to the Author):

Although authors have provided more data, the conclusions remain the same and no more insights on the importance of SETX on the hypoxia-induced replication stress is provided. Reading the abstract one gather that "the biological response to hypoxia includes the accumulation of R-loops and the induction of the RNA/DNA helicase SETX. In the absence of hypoxia-induced SETX, R-loop levels increase, DNA damage accumulates, and DNA replication rates decrease. Thus, SETX plays a key role in protecting cells from DNA damage induced during transcription in hypoxia". This is exactly what it is known for SETX in cells grown in normoxia, so nothing new. But the key question of what so special in SETX expression in hypoxia remains unanswered. The fact that SETX is stress-responsive is a starting observation, but then, where do we go from here? Because SETX also plays a major role in transcription termination, authors failed to provide a clear possible link to this process. Although they show that SETX downregulation in hypoxia leads to changes in gene expression, they did not provide evidence of how this works (e.g.: why a set of genes are upregulated? Is SETX also functioning as a negative regulator of transcription? Is this specific to hypoxia?).

Of note, the description of R-loops resolving factor in the introduction is poor and needs to be expanded.

REVIEWERS' COMMENTS

Reviewer #1 (Remarks to the Author):

The additional data has addressed all of my original criticisms and I do not have any other concerns.

Many thanks to all the reviewers for their time.

Reviewer #2 (Remarks to the Author):

I appreciate the new data and the difficulties to carry out some experiments during COVID. However to link senataxin to hypoxia and ER stress - despite the new data, there are logical leaps that were made. At this point, it is up to the editors on whether this manuscript meet the minimum threshold for publication.

Thank you to the reviewer for the time spent on our manuscript.

Reviewer #3 (Remarks to the Author):

Although authors have provided more data, the conclusions remain the same and no more insights on the importance of SETX on the hypoxia-induced replication stress is provided. Reading the abstract one gather that “the biological response to hypoxia includes the accumulation of R-loops and the induction of the RNA/DNA helicase SETX. In the absence of hypoxia-induced SETX, R-loop levels increase, DNA damage accumulates, and DNA replication rates decrease. Thus, SETX plays a key role in protecting cells from DNA damage induced during transcription in hypoxia”. This is exactly what it is known for SETX in cells grown in normoxia, so nothing new. But the key question of what so special in SETX expression in hypoxia remains unanswered. The fact that SETX is stress-responsive is a starting observation, but then, where do we go from here? Because SETX also plays a major role in transcription termination, authors failed to provide a clear possible link to this process. Although they show that SETX downregulation in hypoxia leads to changes in gene expression, they did not provide evidence of how this works (e.g.: why a set of genes are upregulated? Is SETX also functioning as a negative regulator of transcription? Is this specific to hypoxia?).

We appreciate the reviewers continued interest in our work and agree that the questions raised are all indeed fascinating areas for further work. The reviewer has highlighted the real novelty and interest of our study to be the stress responsiveness of SETX and links to the UPR, we are confident that publication of our manuscript we ensure that many of the questions raised will be addressed by the wider field in addition to ourselves.

Of note, the description of R-loops resolving factor in the in introduction is poor and needs to be expanded.

We agree that this section could be expanded, in the interests of space we have chosen to clearly highlight recent reviews from the Legube, Cimprich and Aguilera labs, all of whom are recognised experts in the field, as a source of additional information for any readers who require it. In addition, we have re-written this section of the introduction to provide more accurate information (bottom of page 3)